# Evolution of chemosensory tissues and cells across ecologically diverse *Drosophilids*

**Gwénaëlle Bontonou** [1,2,5] ✉, **Bastien Saint-Leandre** [1,2,5] ✉, **Tane Kafle**[1,2], **Tess Baticle**[1], **Afrah Hassan**[1], **Juan Antonio Sánchez-Alcañiz**[3] & **J. Roman Arguello** [1,2,4] ✉

Chemosensory tissues exhibit significant between-species variability, yet the evolution of gene expression and cell types underlying this diversity remain poorly understood. To address these questions, we conducted transcriptomic analyses of five chemosensory tissues from six *Drosophila* species and integrated the findings with single-cell datasets. While stabilizing selection predominantly shapes chemosensory transcriptomes, thousands of genes in each tissue have evolved expression differences. Genes that have changed expression in one tissue have often changed in multiple other tissues but at different past epochs and are more likely to be cell type-specific than unchanged genes. Notably, chemosensory-related genes have undergone widespread expression changes, with numerous species-specific gains/losses including novel chemoreceptors expression patterns. Sex differences are also pervasive, including a *D. melanogaster*-specific excess of male-biased expression in sensory and muscle cells in its forelegs. Together, our analyses provide new insights for understanding evolutionary changes in chemosensory tissues at both global and individual gene levels.

Animal's abilities to perceive their chemical environments are remarkably variable. Chemosensory receptor protein families and the cell types in which they are expressed have multiple evolutionary origins[1–6], and the tissues that contain them can differ dramatically across species in morphology and anatomy[7–12]. For example, while taste perception in mammals is largely restricted to gustatory cells located in the mouth, and primarily the tongue, aquatic vertebrates have taste cells distributed externally on their skin[13–17]. Insects have evolutionarily distinct taste receptors and cells that are also broadly distributed across their bodies, including their mouth parts, legs, wings, and ovipositors[18]. Appendages involved in smell are generally more restricted to animals' heads but also differ dramatically among taxa, as exemplified by the bulbous nose of the Proboscis Monkey or the feathery antennae of moths. In addition to differences among species, striking evolutionary changes have also arisen between sexes within species. Sexual dimorphisms in chemosensory perception and organ morphology often evolve rapidly and have been attributed to differences in sex-specific physiological states, sexual selection, and sex-specific nutritional needs, among other factors[19–22].

Understanding the molecular basis of chemosensory evolution is important for fundamental and applied biology. Insights into the genes and expression changes that underlie species' chemosensory differences help us understand how nervous systems adapt in response to varying ecologies and provide the basis for managing disease vectors and agricultural pests. For instance, research on insect chemosensation has advanced our understanding of how mosquitoes track human odors, with important implications for human health[23–25], and has aided in the development of novel farming methods that reduce crop infestation[26]. While these applications draw on knowledge of chemosensation from a broad range of

[1]Department of Ecology & Evolution, Faculty of Biology and Medicine, University of Lausanne, Lausanne, Switzerland. [2]Swiss Institute of Bioinformatics, Lausanne, Switzerland. [3]Instituto de Neurociencias, UMH & CSIC, San Juan de Alicante, Spain. [4]Present address: School of Biological and Behavioural Sciences, Queen Mary University of London, London, UK. [5]These authors contributed equally: Gwénaëlle Bontonou, Bastien Saint-Leandre. ✉e-mail: gwenaelle.bontonou@unil.ch; bastien.saint-leandre@unil.ch; roman.arguello@unil.ch

biological models, much of what we know derives from research on *Drosophila melanogaster*.

Research on *D. melanogaster* has led to extensive knowledge about the development of its nervous system and chemosensory appendages and has generated a nearly complete mapping of its full set of olfactory and gustatory receptor proteins to specific neuron populations. This work has provided the basis for many pioneering functional and behavioral studies[27–32]. In addition, advances in connectomics and single-cell transcriptomics applied to *D. melanogaster*'s nervous system are helping to identify new developmental factors, describe cellular diversity in chemosensory tissues, and characterize synapse-level connections from the peripheral chemosensory neurons to the central brain[33–41]. Beyond its role as a preeminent model for chemosensory biology, *D. melanogaster* and its closely-related species have also long been a model system for evolutionary genetics and speciation[42–47]. The phylogenetic relationships among the *D. melanogaster* species group are well-resolved and include lineages of diverse ages and ecologies. This system, therefore, provides a valuable opportunity to ask how evolutionary forces and environments shape chemosensory systems[22,48,49]. However, beyond the meticulous molecular and cellular characterization of *D. melanogaster*'s chemosensory tissues, little is known about how they evolve between species.

To address this question, we have carried out a comparative transcriptomic experiment in which we generated bulk RNA-sequencing (RNA-seq) datasets for five chemosensory tissues: larva head (mixed sex), ovipositor (female), forelegs (female and male), antennae (female and male), and proboscis with maxillary palps (female and male). These samples were collected from six ecologically diverse species in the *D. melanogaster* species group that share common ancestors between ~0.25–15 million years ago[44,50–53]: *D. melanogaster*, *D. sechellia*, *D. simulans*, *D. santomea*, *D. erecta*, and *D. suzukii* (Fig. 1A). *D. sechellia*, is endemic to the Seychelles island and an extreme specialist on the fruit of *Morinda citrifolia*, which is toxic to the other species[54]. *D. santomea* is endemic to the island of São Tomé and adapted to high-elevation mist forests[55,56]. *D. erecta* is restricted to west-central Africa and is thought to be an opportunistic specialist on the fruits of *Pandanus*[57]. *D. suzukii* originated in Eastern Asia but has expanded rapidly worldwide in the last decade[58,59]. Unlike the other species, *D. suzukii* females oviposit in ripe soft-bodied fruits and, as a result, have become a global agricultural pest[60–65]. Both *D. simulans* and *D. melanogaster* are generalists that feed on a broad range of decaying fruits and have nearly worldwide distributions[66]. We found that stabilizing selection has been the predominant force shaping the evolution of chemosensory transcriptomes. Still, several thousand genes have evolved expression changes in each tissue. Intriguingly, genes that have changed expression in one tissue have usually changed expression in multiple other tissues but at different times in the past, suggesting widespread tissue-specific regulatory changes. The fast evolution of chemosensory gene families and sex differences are also prominent and highlight distinct ecological differences. These data can be explored with our dashboard available at: https://ctct.unil.ch/.

## Results

### Relationships between sensory tissue transcriptomes
To study the evolution of gene expression in the main chemosensory tissues of closely-related *Drosophila*, we generated bulk RNA-seq datasets for six ecologically diverse species and five sensory tissues (Fig. 1A; Methods). On average, we obtained 43 million mapped reads per sample with high correlations across triplicates (average Pearson correlation coefficient = 0.98). To overcome annotation biases, we used these datasets to produce equivalent de novo gene annotations and used the resulting gene sets for orthology/paralogy assignments. This approach resulted in similar genome annotations with BUSCO scores ranging from 91.9–97.3%, indicating a well-balanced dataset for cross-species comparisons.

We began investigating the relationship between chemosensory tissue transcriptomes by conducting a principal component (PC) analysis on expression levels of 12,096 genes with a one-to-one relationship across the six species (1:1 orthologs; Fig. 1B). The first principal component (PC1) separates the three appendage samples from the larval head and ovipositor samples. The genes that contribute the most to the negative loading of PC1 are enriched for gene ontology (GO) terms related to cilia, cell projections/axons, and synapses, among other neural categories. These terms contrast with the enrichment of cell cycle, organelle, and nucleus-related terms that most contribute to the top positive loadings of PC1 (Supplementary Fig. 1). Soft-clustering analysis of correlated expression changes across multiple genes also identified appendage-specific expression modules that are enriched for cilium, dendrite, and chemosensory terms that load negatively on PC1 and larval/ovipositor-specific modules that are enriched for cell cycle ontology terms that load positively on PC1 (Supplementary Fig. 1). The second principal component (PC2) separates the antenna from the other samples and is enriched for GO terms related to olfactory, dendrite, and sensory function for the top positive loadings. We again identified an antenna-specific module that is enriched for olfactory receptor, odorant binding, and dendrite terms that load positively on PC2 (Supplementary Fig. 1). The gene set defining this antenna module negatively correlates with a muscle-related gene module that is enriched in the forelegs and proboscis datasets (Supplementary Fig. 1), highlighting both neural and structural genes underlying the chemosensory tissue transcriptome differences.

We observed further separation between the antenna, ovipositor, and larva clusters with additional PC pairings, but the foreleg and proboscis+palps transcriptomes always overlap (Supplementary Fig. 2). The latter overlap is likely driven primarily by the foreleg and proboscis tissues, given relatively small tissue contribution of the maxillary palps. Among the five tissues, the ovipositor samples varied the most in PC space, reflecting the lower correlation across some replicates (Methods). Despite this variation, the clustering separated the *D. suzukii* samples, for which the replicates were highly correlated (average Pearson correlation coefficient = 0.97). This *D. suzukii* difference is notable because the females of this species differ from the others in their preference for ovipositing in ripening fruits (instead of overripe/rotting fruits) and have evolved an elongated serrated ovipositor that punctures fruit skins[67].

### Sensory transcriptomes exhibit low rates of divergence, with a few exceptions
To investigate the clustering of the transcriptomic datasets on a species level, we estimated expression distances by applying an evolutionary model of transcriptome divergence[68]. The clustering largely recapitulates the known phylogeny (Fig. 1C). For all tissues except the larva head, the consistent difference between the species' genetic relationships and the transcriptomic clustering is the lack of an internal node shared by *D. erecta* and *D. santomea*. The transcriptomic clustering of the larval head dataset results in additional discrepancies, with *D. erecta* grouping with *D. melanogaster* and *D. simulans*, *D. sechellia* and *D. santomea* grouping together (Fig. 1C). This pattern points to a more complex evolutionary history of gene expression evolution for the larva head compared to the other tissues.

The distinct ecologies and evolutionary histories among these six species led us to hypothesize that their chemosensory transcriptomes have evolved at different rates. We tested for these differences by applying relative rate tests, which use a pair of ingroup species with an outgroup species to determine whether one of the two ingroup lineages has a significantly elevated rate of transcriptomic change. We applied this test to all 12,096 1:1 orthologs for all species-pairs (setting *D. suzukii* as the outgroup) and found that the distribution of test scores (Z-scores) for the majority of the species-tissue-sex

comparisons are largely consistent with equal rates of transcriptomic change across species, indicating that sensory transcriptomes exhibit low rates of divergence (Fig. 1D). We obtained consistent results when examining the distribution of Z-scores based on subsampled sets of the 1:1 orthologs (Fig. 1D) and when using either *D. erecta* or *D. santomea* as outgroup species to *D. simulans*, *D. sechellia*, and *D. melanogaster* (Supplementary Fig. 3).

Although most tissue's transcriptomic divergence was low, we identified several tissues that stand out with elevated species-specific and sex-specific differences. *D. simulans* has a significantly elevated rate of evolution for its female antenna transcriptome (Wilcoxon signed-rank test V = 4.6e + 06, *p* < 0.001), larva head transcriptome (Wilcoxon signed-rank test V = 7.1e + 06, *p* < 0.001), and ovipositor transcriptome (Wilcoxon signed-rank test V = 2.9e + 06, *p* < 0.001). In

addition, *D. melanogaster's* male forelegs and ovipositor transcriptomes have significantly elevated rates of transcriptome change (Wilcoxon signed-rank test V = 6.6e + 06, *p* < 0.001 *and* V = 3.1e + 06, *p* < 0.001, respectively). In contrast, the ovipositor transcriptomes from *D. santomea* and *D. erecta* were both found to have significantly lower expression divergence compared to the other species (Wilcoxon signed-rank test V = 3.1e + 06, *p* < 0.001 and V = 2.3e + 06, *p* < 0.001, respectively). Overall, these global analyses of transcriptomic differences highlight a limited set of sensory tissues as rapidly evolving among the species, possibly reflecting key ecological and/or functional differences. They also provide evidence for significant sex differences that exist within species (see below).

The observation that the chemosensory transcriptomes generally display low rates of divergence is suggestive of stabilizing selection but

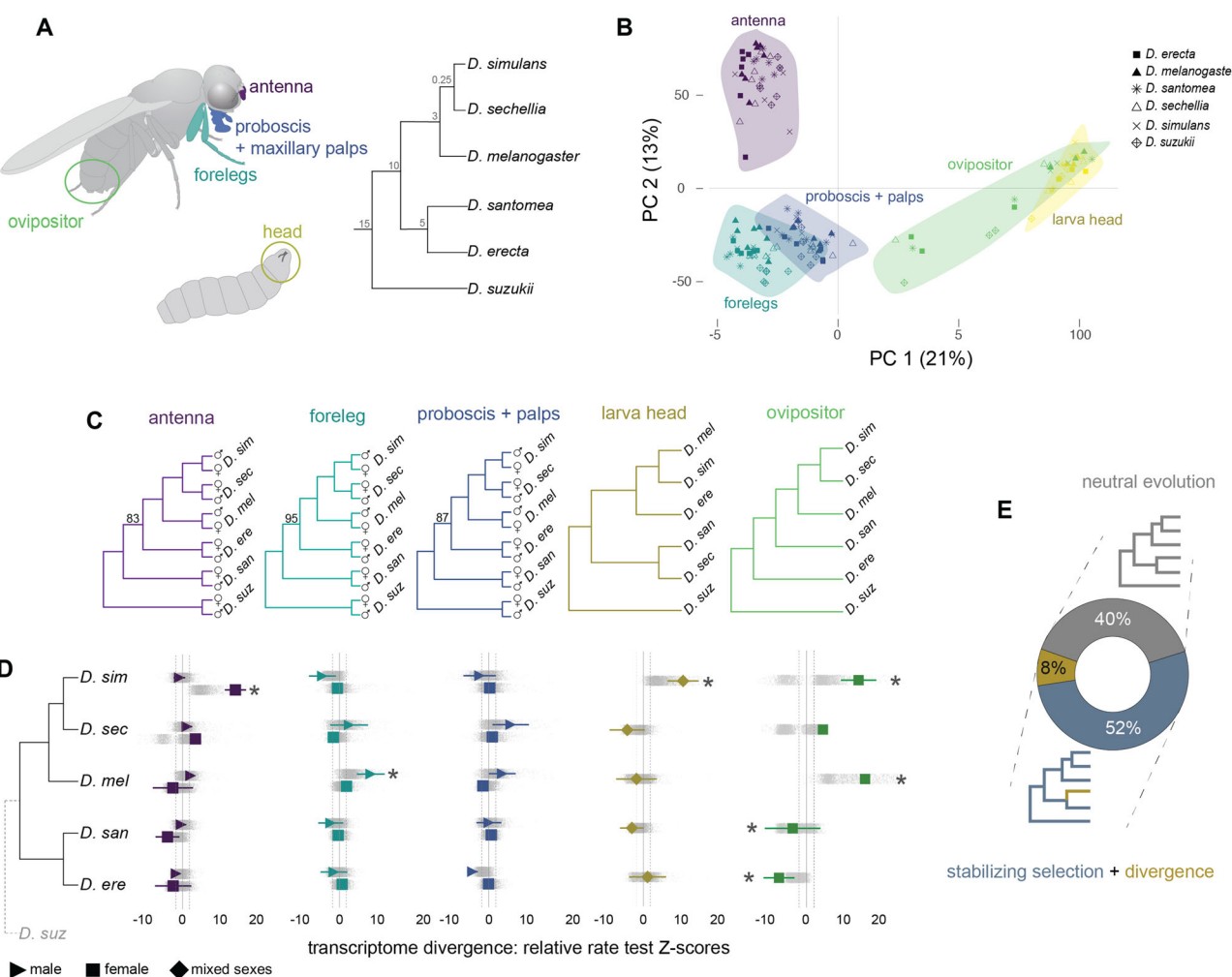

**Fig. 1 | Chemosensory tissue transcriptome evolution. A** Overview of the chemosensory tissues and species used in this study. The numbers at the nodes of the species tree are the estimated divergence dates in millions of years. Antennae, proboscis+maxillary palps, forelegs, ovipositors and larvae transcriptomic datasets are labeled in purple, blue, cyan, green and dark yellow, respectively. **B** PCA of the transcriptomic datasets using 12,096 1:1 orthologs. The percentages on the axes are the amount of variation explained by the PCs. Each dot corresponds to one dataset. **C** Clustering of the transcriptomic datasets (12,096 1:1 orthologs) according to species and sex. Numbers above branches are bootstrap values for nodes with support <100. Species names are abbreviated to the first three letters. **D** Relative rate test results arranged by the species' phylogeny. Tissues are arranged to align vertically with **C**. Colored shapes and lines display the mean and standard deviation of Z-scores from the full set of 1:1 orthologs. *D. suzukii*, noted with the dashed line and gray font, was used as the outgroup species. Gray data points are Z-scores that

resulted from repeating the tests with subsampled datasets (Methods). Asterisks denote the significantly elevated (positive values) or reduced (negative values) rates of gene expression change (Wilcoxon test comparing Z-score distribution to the minimum and maximum values of non-significant Z-scores: dotted lines). Only significant one sample right-tailed Wilcoxon tests are displayed (*D. simulans* female antennae, *n* = 4000, *p* < 2.2⁻¹⁶; *D. melanogaster* male legs, *n* = 4000, *p* = 3.73⁻²⁸⁸; *D. simulans* larva, *n* = 4000, *p* < 2.2⁻¹⁶; *D. simulans* ovipositor, *n* = 4000, *p* = 9.65⁻⁷; *D. melanogaster* ovipositor, *n* = 4000, *p* < 2.2⁻¹⁶) as well as significant one sample left-tailed Wilcoxon tests (*D. santomea* ovipositor, *n* = 4000, *p* = 2.38⁻³³; *D. erecta* ovipositor, *n* = 4000, *p* = 6⁻¹²⁵). Species names are abbreviated to the first three letters. **E** Fraction of expression tree branches (for all 1:1 orthologs and all tissues) found to best fit one of three evolutionary models: neutrally evolving, constrained, or divergent. Note that the divergent branches were found within otherwise constrained gene trees. Location of source data for this figure can be found in "Source_data.xlsx".

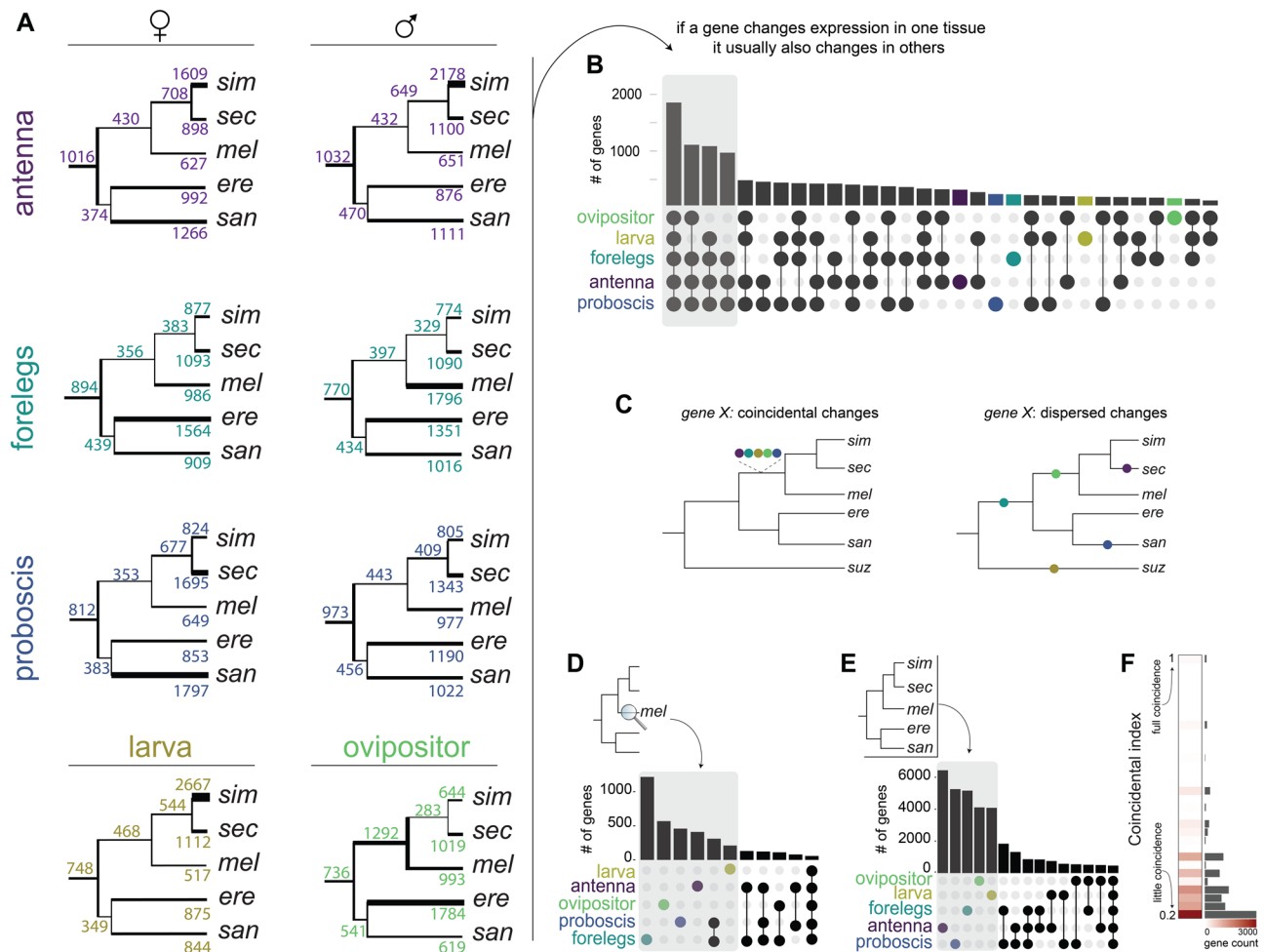

**Fig. 2 | Expression changes over branches and tissues. A** Expression changes inferred across species for the five tissues and sexes. The number above each branch is the total number of expression changes (up and down), and the thickness of the branch is proportional to that number. **B** Quantification of the genes that change in expression across multiple tissues. The height of each bar indicates the number of genes that have changed expression across the set of tissues indicated by the darkened/colored circles. **C** Schematic illustrating a hypothetical *gene X* having expression changes involving multiple tissues that were coincidental (occurring on a single branch) or dispersed (occurring across multiple branches). In this illustration, the color of the dot indicates the tissues in which *gene X*'s expression changed. **D** Quantification of the coincidental expression changes along the branch leading to *D. melanogaster*. The height of each bar indicates the number

of genes that have changed expression across the set of tissues indicated by the darkened/colored circles. The plot was truncated at the bin containing the overlap of the five tissues. **E** Quantification of the coincidental expression changes over all branches in the phylogeny. The height of each bar indicates the number of genes that have changed expression across the set of tissues indicated by the darkened/colored circles. The plot was truncated at the bin containing the overlap of the five tissues. **F** Summary of the distribution of the "coincidental index" for all expression changes. A coincidental index of 1 indicates that a gene has changed in expression in all tissues within a single branch (full coincidence), decreasing values indicating less coincidence. Location of source data for this figure can be found in "Source_data.xlsx".

could also result from similar rates of neutral evolution over the relatively short timespans. We thus quantified the relative contributions of genetic drift and stabilizing selection to the evolution of gene expression. We did this by testing if the data for each gene expression tree is better explained by a phylogenetically-based model of neutral expression evolution or a model that assumes evolutionary constraint (with one or more branches experiencing expression divergence (Methods)). Across all gene expression trees, 52% of branches were inferred to have been under selective constraint with 8% of branches inferred to have diverged within an otherwise constrained expression tree. Approximately 40% of the branches display evidence of neutrally evolving expression (Fig. 1E).

## Genes change expression in multiple tissues but at different evolutionary times

As illustrated by the outliers in the relative rate tests and the detection of divergent branches in constrained expression trees, broad

selective constraint has not precluded individual genes from evolving species-specific differences. Therefore, we investigated the genes that have changed in expression between species and when in the past the changes took place. Using phylogenetically-informed tests applied to our set of 1:1 orthologs, we detected several thousand differentially expressed genes for each tissue. Most of these expression changes occurred in only one species. The total number of expression changes ranges from 8499 (involving 6697 of the 1:1 orthologs) in male antennae to 7501 (involving 5927 of the 1:1 orthologs) in female legs (Fig. 2A, Supplementary Data 1). Analysis of the functional categories enriched by these differentially expressed genes highlighted combinations of developmental/morphological, neural/sensory, and gene regulation terms, among others, in varying proportions along extant and past lineages (Supplementary Fig. 4, Supplementary Data 2). The elevated number of expression changes identified in *D. simulans* female antenna (1609) and larva (2667), and in *D. melanogaster* male forelegs (1796), confirms a history of

elevated rates of expression evolution for these tissues (Fig. 2A; see also Fig. 1D).

Having identified gene expression changes across species for each tissue on its own, we next questioned how often a given gene changed expression in multiple tissues. Quantifying these overlaps revealed that genes that have changed expression in only one tissue are rare (~7%). Instead, most genes have changed in expression across multiple tissues, with the set of genes displaying changes across all five tissues being the largest set by almost twofold (Fig. 2B). Importantly, we find similar results for tissue overlaps and functional category enrichment when identifying differentially expressed genes using a standard alternative (non-phylogenetic) approach, confirming the robustness of our findings (Supplementary Fig. 5; Methods).

When genes have changed their expression across multiple tissues, this could have occurred simultaneously (e.g., as a result of pleiotropic mutations) or it could have resulted from the accumulation of tissue-specific expression changes at dispersed times in the past (e.g., as a result of the evolution of *cis*-acting regulators or of changes in cell abundances) (Fig. 2C). To gauge the importance of these two contrasting possibilities, we estimated the number of times a gene changed in expression across multiple tissues on individual branches of the phylogeny. Our analysis revealed very few coincidental changes. For example, on the branch leading to *D. melanogaster*, a vast majority of the expression changes occurred in only one of the five tissues (Fig. 2D). The same trend holds when summarizing expression changes over all branches of the phylogeny (Fig. 2E), as well as when quantifying the rate of coincidental changes (Fig. 2F). Collectively, these analyses imply that most differentially expressed genes have evolved expression changes across different tissues at independent times in the past, consistent with independent evolutionary changes in gene regulation and/or cellular abundances.

Further inspection of the rare coincidental expression changes indicated that the probability of their occurrence is independent of branch lengths (Supplementary Fig. 6A). This finding confirms the intuition that many of these expression changes have arisen by pleiotropic mutations (and are not primarily a result of low resolution for detecting independent expression changes along longer branches). Interestingly, the most frequent coincidental change among all tissue combinations involved the forelegs and proboscis + palps samples (Supplementary Fig. 6B; see also Fig. 2D, E). This observation is coherent with the transcriptomes of these two tissues being the most similar among the five (Fig. 1B) and points to the likelihood that they share gene regulatory networks.

### Evolution of gene expression is often cell-type specific

Our finding that most differentially expressed genes across species are expressed in all or many of the tissues (Fig. 2B) led us to question if their expression specificity differs from genes that have not changed across species. We first compared the specificity of expression between differentially and non-differentially expressed genes at the tissue level and found that genes that have changed in expression have similar modes of tissue specificity but tend to be somewhat more tissue-restricted than genes that have not changed (Fig. 3A, B; Wilcoxon signed-rank test $V = 4.3e + 10$, $p < 0.001$). We then asked a similar question but at the level of cell types instead of tissues. Using the recently generated *D. melanogaster* single-cell atlases for antenna, legs, and proboscis[33] (Fig. 3A), we measured expression specificity at the level of cell types. We found that differentially expressed genes are significantly more likely to be expressed in a limited number of cell types than genes that have not changed in expression across species (Fig. 3B).

The relationship between tissue specificity and cell specificity varies substantially. For example, we identified differentially expressed genes that are expressed narrowly at the cell and tissue levels, e.g., the olfactory receptor *Or56a*, a receptor used by *Drosophilds* to detect the harmful mold odor geosmin[69] (Fig. 3B). In contrast, we also identified genes that are expressed intermediately at the tissue level but are highly cell-specific within tissues. Using previous cell annotations[33] and marker-based cell type identification across the three atlases, we verified that these latter cases can be attributed to the same cell types being shared across tissues, e.g., *sosie*, a membrane protein localizing to mechanosensory cells and *rho*, a serine protease that localizes to glial cells (Fig. 3B). These examples illustrate how measurements of expression specificity using bulk tissues can mask the cell specificity of a gene's expression[70]. They also demonstrate that species' expression differences that likely underlie phenotypic divergence can be ascribed to individual cell populations.

### New genes tend to be cell-specific

New genes are a key source of evolutionary novelty[71]. Due to their potential contributions to species differences, we expanded our analyses to examine how gene age and duplication frequency relate to differences in the transcriptomes of sensory tissues. We compared the specificity of tissue expression between old genes (genes that predated the diversification of the *Drosophila* subgenus more than 50 million years ago) and new genes (genes that arose since). Consistent with previous work[72–76], we found that new genes are significantly more likely to be expressed in fewer tissues than old genes ($p < 0.001$; Fig. 3C). We also found that the more often a gene has been duplicated, the more tissue-restricted its paralogs are (Fig. 3C). We reasoned that the increased expression specificity of new genes would translate to their detection in a narrower number of cell types. We mapped the expression of new genes to the single-cell atlases for the antenna, legs, and proboscis and compared their expression specificity across cell types to that of old genes. Our analysis confirmed that new genes are indeed significantly more likely to be cell type specific than old genes (Fig. 3C; Wilcoxon signed-rank test $V = 3.47e + 07$, $p < 0.001$).

### Pervasive expression evolution of chemosensory genes

Insect genomes contain three large chemoreceptor gene families: odorant receptors (*Or*s), gustatory receptors (*Gr*s), and ionotropic receptors (*Ir*s)[27]. In addition, members of the chemosensory protein family (*CSP*s), and other diverse protein families, including the odor binding proteins (*Obp*s), transient receptor potential channels (*Trp*s), and pickpocket ion channels (*ppk*s), are chemoreceptors or otherwise involved in the peripheral sensing of environmental chemicals[27,77–79]. The patterns of expression for most of these "chemosensory genes" have been mapped to specific tissues and cell populations in *D. melanogaster* and have provided the foundation for numerous functional and behavioral studies[27]. While multiple RNA-seq experiments have detected expression differences among developmental stages or species (or both) for chemosensory genes[20,80–83], the heterogeneous combination of samples, experimental design, and sequencing approaches have limited evolutionary analyses. We, therefore, manually curated a set of 368 chemoreceptor genes for the above seven gene families and used our uniformly generated RNA-seq dataset to investigate how their expression patterns have evolved between species, tissues, and developmental stages (Supplementary Data 3).

Out of the 368 chemosensory genes, we detected expression for 299 in at least one of the tissues. In the antenna, proboscis, and forelegs samples, the expression patterns largely matched previous reports[20,32,80,81,83–111] (Supplementary Data 4). However, we detected few of the described *D. melanogaster* chemosensory genes in larva head samples, likely due to their very low expression levels. In each tissue, we identified a core set of genes that were expressed across all six species (antenna = 98, proboscis = 71, forelegs = 63, ovipositor = 28, larva head = 37; Supplementary Data 5). Fourteen chemosensory genes were found to be expressed across all tissues, including two members of the *ppk* family, *ppk* and *ppk26*, which have previously been implicated in the detection of noxious mechanical stimuli in larvae. Their

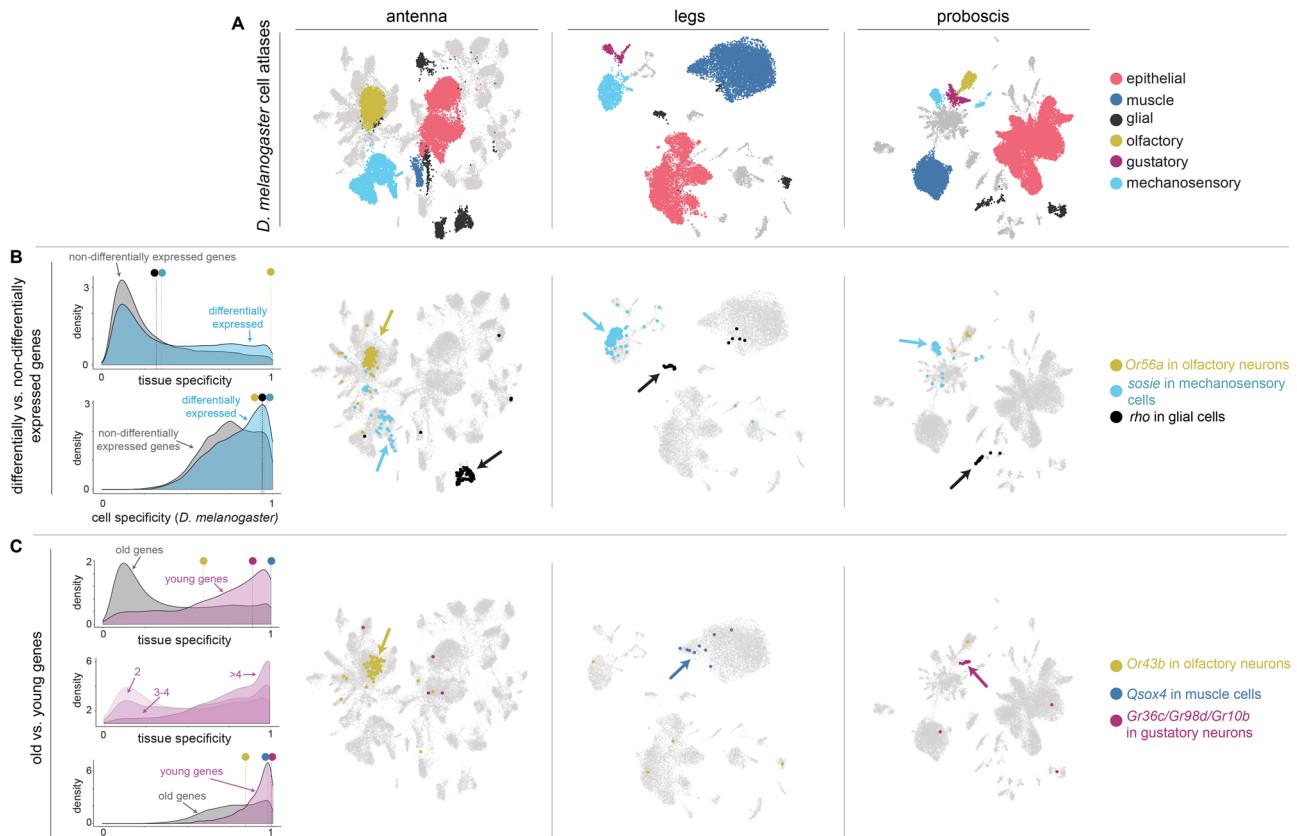

**Fig. 3 | Specificity of gene expression at the level of tissues and cells. A** Single cell atlases from *D. melanogaster* antenna legs and proboscis. Colors highlight the same cell types of interest across tissues. **B** Density plots for differentially and non-differentially expressed genes (leftmost panel) relative to expression specificity for tissues (top) and cell types (bottom). Colored circles with lines above the density plots indicate the expression specificity values of three genes (*Or56a*, *sosie* and *rho*) chosen to illustrate the varying relationships between expression specificity at the level of tissues and cells. Expression of the three genes has been mapped onto the *D. melanogaster* cell atlases (right three panels). **C** Density plots for old and young genes relative to expression specificity for tissues (top left) and cell types (bottom left). The middle left density plot shows the distribution of expression specificity values for genes grouped by duplication levels (2 = paralog group size of 2, 3–4 = paralog group size of 3–4, >4 = paralog group size greater than 4). Colored circles with lines above the density plots indicate the expression specificity values of five genes (*Or43b*, *Qsox4* and *Gr36c/Gr98d/Gr10b*) chosen to illustrate varying relationships between expression specificity at the level of tissues and cells (see text). Expression of the five genes has been mapped onto the *D. melanogaster* cell atlases (right three panels). Location of source data for this figure can be found in "Source_data.xlsx".

broad expression suggests additional sensory roles for these proteins in adults. The detection of multiple *Che* members in each is also notable, given that their suspected roles in detecting contact pheromones and pathogens have hitherto been limited to the legs[98,112,113].

When we screened the set of 1:1 orthologous chemosensory genes for differential expression, we found that nearly all of them have evolved expression changes in at least one branch of the species tree (Fig. 4A, 93% of *CSP*s, 96% of *Gr*s, 100% of *Ir*s, 100% of *Ppk*s, 100% of *Obp*s, 98% of *Or*s, 100% of *Trp*s; Supplementary Data 6). Furthermore, most genes have experienced recurrent expression changes, with the *CSP* family showing the greatest number. Consistent with genome-wide patterns (Fig. 2A), most expression changes were species-specific. Among the differentially expressed chemoreceptor genes, those that have gained or lost expression in a particular tissue were of particular interest because they may indicate novel gains (or losses) of sensory capabilities. We defined a gene with an average transcript per million (TPM) greater than 3 as expressed and genes with an average TPM less than 0.5 as unexpressed. Using these thresholds, we identified 95 chemosensory orthologs (32%) that have either gained or lost expression in at least one tissue. Some of these expression gains/losses have occurred once, as illustrated by the gain of expression of *Gr98a* and *Gr98b* in *D. melanogaster's* ovipositor or *Gr59e* in *D. erecta's* larva, while others have involved recurrent changes, as for the foreleg-

expressed *CheB74a* or the antenna-expressed *Ir31a* (Supplementary Figs. 7–13). Similar analyses of the set of 85 chemosensory genes (from 31 gene families) that have duplicated since the common ancestor of the six species revealed that nearly all recent paralogs have retained expression in the same tissues but often at lower levels (Supplementary Figs. 14–15, Supplementary Data 7). The two exceptions are *Gr59a4* and *Ir52f2* in *D. suzukii*, which show gains of expression that may indicate the neofunctionalization of these genes (Supplementary Fig. 16).

To gain spatial and cellular resolution for the expression of a subset of 95 chemosensory genes with novel species-specific expression patterns, we designed in situ hybridization chain reaction (HCR) experiments for six of them: *Gr32a*, *Gr33a*, *Gr61a*, *Ir7f*, *Or1a*, *Or45a* (Methods). We detected expression that was consistent with our RNA-seq results for all of these genes except *Gr32a* (Fig. 4B–F, Supplementary Fig. 17). For unknown reasons, we were unable to detect *Gr32a* in *D. suzukii* antenna despite detecting its expected expression in the labial palps (Supplementary Fig. 18). Additional co-labeling experiments using probes for other cell type markers resulted in the discovery of unexpected patterns of cellular expression including for the two gustatory receptors, *Gr33a* and *Gr61a*. In *D. melanogaster*, Gr33a is characterized as a bitter receptor expressed in taste cells in the legs and proboscis and involved in aversion to male-male courtship[102,114,115]

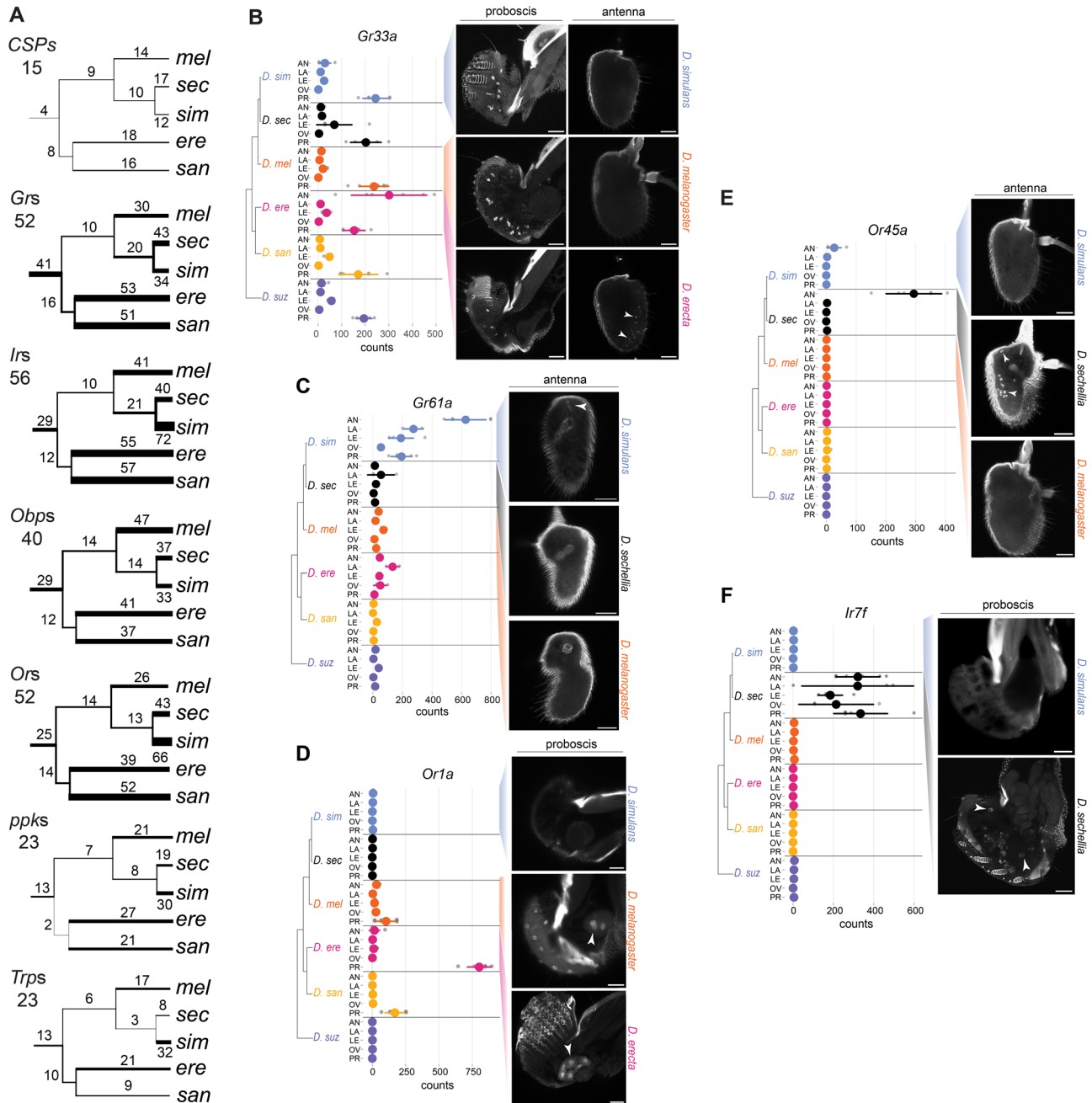

**Fig. 4 | Evolution of chemosensory gene expression. A** Expression changes mapped onto the species tree for genes belonging to the main chemosensory gene families (*Gr*s gustatory receptors, *Ir*s ionotropic receptors, *Or*s odorant receptors, *CSP*s chemosensory proteins including *CheA* and *CheB* family members, *Obp*s odorant binding proteins, *ppk* pickpocket ion channels, *Trp*s transient receptor potential channels). The number above each branch is the total number of expression changes (up and down) across tissue samples, with the thickness of the branch proportional to that count. The number under the gene family name corresponds to the number of 1:1 orthologs used for the analysis. **B**–**F** in situ HCR results for chemosensory genes with a species-specific gain of expression. On the left is the species tree (not to scale) with the mean normalized read counts obtained for each sample. The mean and standard error are represented by the colored dots and the vertical line, respectively, with the individual data points in gray. AN antenna, LA larva head, LE forelegs, OV ovipositor, PR proboscis. RNA in situ hybridizations on the right with the targeted tissues(s) above each column. White arrows indicate cells with species-specific receptor expression. Scale bars: 30 μm. See also Supplementary Figs. 17–18. In **C** and **E**, autofluorescence highlights the sacculus in the three species. Location of source data for this figure can be found in "Source_data.xlsx".

We found that expression of *Gr33a* has expanded from bitter taste neurons into olfactory sensory neurons (*Orco*+) in the antenna of *D. erecta* (Fig. 4B, Supplementary Fig. 17A). Interestingly, antennal expression of *Gr33a* was previously observed in *D. melanogaster* when programmed cell death was experimentally blocked in olfactory sensory neurons[116], possibly indicating a *D. erecta*-specific developmental "escape" from cell death for this neuron population. Analogously,

*Gr61a*, a glucose receptor in *D. melanogaster* that is expressed in neurons in the labellum, legs, and the labral sense organ[117,118], was also found to have expanded into olfactory neurons of *D. simulans*' antenna (Fig. 4C, Supplementary Fig. 17B). Analyses of two odorant receptors, *Or1a* and *Or45a*, also revealed species-specific expression patterns. We found that *Or1a*, which was previously described as being larva-specific in *D. melanogaster*[104], is expressed in non-neuronal cells that are likely

part of the labellar glands in *D. melanogaster* and *D. santomea* (Fig. 4D, Supplementary Fig. 17C). To our knowledge, no chemosensory function for this gland has yet been described. We found that *Or45a*, previously described as larva-specific in *D. melanogaster*[109], is also expressed in the adult antenna in *D. sechellia* (Fig. 4E, Supplementary Fig. 17D). Finally, *Ir7f*, which has yet to be functionally characterized, was one of the most distinct differently expressed chemosensory genes because it has uniquely gained high expression in all chemosensory tissues in *D. sechellia* (an example of a "coincidental" gain of expression). We observed expression of *Ir7f* within cells that also express a pan-neuronal marker (*nsyb*) in the labial palps, indicating that this gene likely encodes a taste receptor (Fig. 4F, Supplementary Fig. 17E). Together, these expression analyses underscore the remarkable evolutionary flexibility in transcript abundance, developmental timing, and spatial expression of chemosensory genes.

### Fast evolution of sex differences

Next, we identified sex differences in our dataset and placed them in a phylogenetic context. *Drosophila* chemosensory tissues are involved in sex-specific functions and often vary between the sexes in morphology and neuroanatomy[19,67,119–121]. While previous single gene and transcriptomic analyses identified sex differences in gene expression within some of these tissues[20,81,82,122], their evolutionary histories between tissues and species remain unclear.

For each species, we computed the number of genes with significantly different expression levels between males and females (≥1.5-fold change with adjusted $p < 0.01$) within our proboscis + palps, antenna, and foreleg datasets and examined their variation among the six species. Our analysis revealed extensive evolution in the number of sex-biased genes across species, the proportion of genes having male-versus female-biased expression, and in the identity of the sex-biased genes (Fig. 5A; Supplementary Data 8). Remarkably, the patterns of sex-biased gene expression do not reflect the genetic relationships among the species, in line with previous findings that expression differences between the sexes evolve quickly[123–127]. We observed an approximately ten-fold difference in the number of sex-biased genes between the species with the fewest and the species with the most (*D. santomea* and *D. sechellia* with 135 and 178, respectively, versus *D. erecta* and *D. melanogaster* with 1350 and 1132, respectively). Although the number of male-biased genes outnumbers female-biased genes (2098 vs. 1287), this ratio varied considerably across tissues. Genes expressed in the forelegs and the proboscis are mainly male-biased, while female-biased genes are predominant in antennae. These results suggest that different modes of sexual selection may have shaped the male/female expression balance in a tissue-specific manner.

We then asked if the identity of sex-biased genes is shared across species and tissues. These analyses once again highlighted pervasive variation in the sets of genes that differ between the sexes. In a majority of cases, the sets of female- and male-biased genes are private to each species (Fig. 5B). Intriguingly, among the few overlaps between species, we found enrichments of genes involved in or activated by cell-autonomous and non-autonomous control of sex differences (including *fruitless*, *doublesex*, *insulin-like peptide 7*, and members of the cytochrome P450 family; hypergeometric tests $p < 0.001$), suggesting they may play roles in the maintenance of sexually dimorphic traits in adult sensory tissues, similar to what has been observed for *D. melanogaster*'s intestine[21]. We also observed enrichment in chemosensory proteins among conserved sex-biased genes (hypergeometric tests $p < 0.001$) which have been shown to be sex-biased and involved in pheromone-induced behaviors[98,112]. Finally, within species, if a gene is sex-biased in one tissue, it is rarely sex-biased in the other two (Fig. 5C).

We sought further insight into the cell types underlying the derived *D. melanogaster* male-biased foreleg expression. Of the 806 male-biased genes identified in our RNA-seq experiment, 285 were

detected in the leg samples of the Fly Cell Atlas[33]. Examination of the cell-type-specificity of the 285 genes revealed that most are expressed very narrowly (number of genes with cell specificity >0.8 = 257) and are enriched in cell populations related to mechanosensation, gustation, and muscle (Fig. 5D–F). These cell types are particularly compelling in light of the extensive literature identifying key roles for these sensory modalities in *D. melanogaster*'s courtship[22,128–130] and because musculature is dimorphic among *Drosophilids*[131]. Sexually dimorphic genes in muscle include those with functions in mitochondrial respiratory (*ND-B8*), actin assembly (*forked*), and vision (*Rh2* and *Culd*). Sexually dimorphic genes in the mechanosensation/gustation cell populations include a putative pheromone receptor (*Ir52c*), two *trp* channels involved in temperature sensing (*brv3* and *pkd2*), and several genes involved in neuron development and signaling (e.g., *Unc-104* and *Stathmin*).

Sex-bias expression that is detected in bulk tissue samples could result from differences in cell abundances between the sexes, transcript abundance differences between the sexes, or a combination of both. We examined these possibilities using the sex-specific Fly Cell Atlas leg data, which includes two pooled male and two pooled female samples. Though preliminary due to the limited number of replicates, we found that the population of muscle cells enriched for male-biased genes (identified in the bulk RNA-seq analysis) is more abundant in the male sample compared to the female sample. We also found higher mean expression levels for the male-biased genes (identified in the bulk RNA-seq analysis) in the male sample compared to the female sample (Wilcoxon test $p < 0.01$). No differences in cell abundance or expression levels between the sexes were identified in the mechanosensation/gustation cell population (Supplementary Fig. 19). These results suggest that, at least in the muscle cells, regulation of the cell population size and transcript abundance have both contributed to the sexual dimorphism in *D. melanogaster*. Additional male and female samples, along with cross-species single-cell datasets will provide more rigorous tests of these observations.

## Discussion

By conducting comparative transcriptomic analyses of chemosensory tissues across species and linking them with single-cell datasets and additional in situ FISH experiments, we have expanded our understanding of how these sensory systems evolve on a global and individual gene level. Globally, we have found that stabilizing selection has been the dominant evolutionary force shaping chemosensory transcriptomes, with ~60% of the branches on the 1:1 ortholog expression trees displaying evidence of evolutionary constraint. Because protein abundances are more directly relevant to phenotypic changes than mRNA levels, we suggest there is additional stabilizing selection at the level of translation. Support for this comes from cross-species comparisons of co-profiled translatomes and transcriptomes which have found fewer between-species changes at the level of the translatome compared to the transcriptome, reflecting additional stabilizing selection for protein abundance[132]. Nevertheless, evolutionary constraint has not precluded a subset of tissues and genes from experiencing accelerated rates of expression change. At the transcriptomic level, *D. melanogaster* (forelegs and ovipositor) and *D. simulans* (antenna, larva head, and ovipositor) are distinct for having significantly increased expression divergence. This was initially curious as the two are ecological generalists while the other species have evolved ecological specializations. However, it is consistent with *D. melanogaster* and *D. simulans* having the largest effective population sizes (and likely substantially so)[47,51] resulting in positive selection playing a greater role within these species compared to the others. If true, this result would suggest an important role for positive selection in driving gene expression changes.

At the level of individual genes, we have identified numerous instances of significant expression differences across species for each

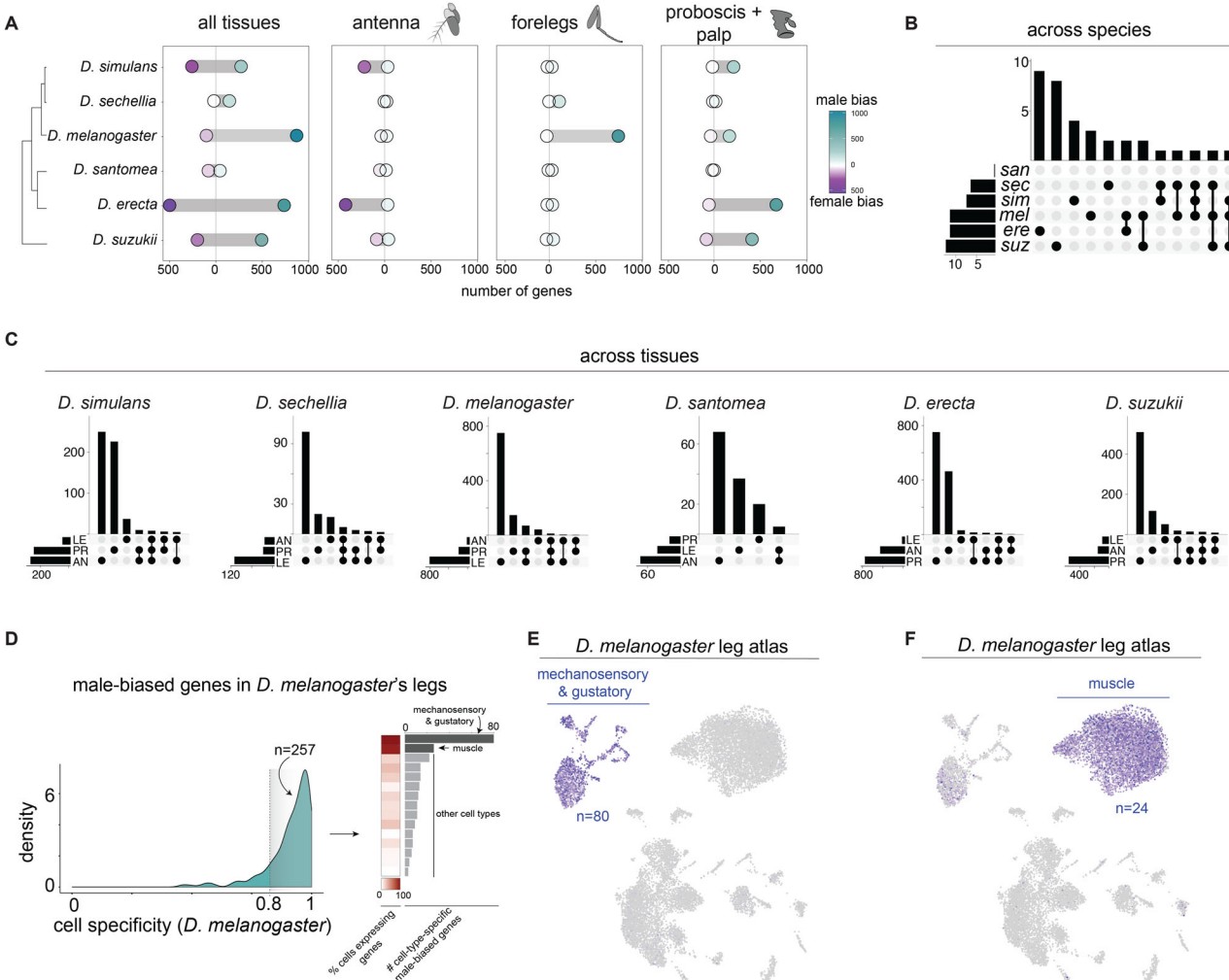

**Fig. 5 | Evolution of sex-biased expression in chemosensory tissues. A** Number of male- and female-biased genes across species and tissues. Sex-biased gene expression does not match the species' phylogenetic relationships demonstrating many species-specific changes. **B** Overlap across species (regardless of tissue) for the number of genes that share the same direction of sex-bias (male or female), illustrating that sex-biased genes are species-specific. Species names are abbreviated to their first three letters. **C** Number of sex-biased genes that overlap across tissues within species. Most sex-biased genes are tissue-specific (LE = forelegs, AN = antenna, PR = proboscis+palps). **D** (left) Density plot for *D. melanogaster's* male-biased genes relative to cell-type specificity. (right) Heat map showing the fraction of cells in a given cell population that express the male-biased genes and bar plot displaying the total number of male-biased genes found expressed within a given cell population. Most male-biased genes are cell type-specific and predominantly found within cells associated with mechanosensation, gustation, and muscles. **E, F** Cell atlas for *D. melanogaster* legs with the total mean expression of 104 male-biased genes displayed, highlighting their restricted expression in mechanosensory/gustatory and muscle cells. Location of source data for this figure can be found in "Source_data.xlsx".

of the five chemosensory tissues. We estimate that ~8% of all expression tree branches display changes consistent with adaptation. A previous estimate based on a "fitness seascape" model of gene expression evolution inferred that ~60% of expression divergence among (partially overlapping) *Drosophila* species was driven by positive selection, concluding that adaptation is the principle evolutionary force underlying the changes[133]. Though seemingly at odds with our observations, we propose that this previous estimate is comparable to ours if we summarize our results at the level of genes instead of trees. If we summarize the percentage of gene expression trees that have at least one divergent branch, we find that ~50% have experienced historical change(s) in expression optima that may be consistent with adaptation (Supplementary Fig. 20). However, a gene-focused summary obscures our finding that most branches in the expression trees are under constraint. It is also noteworthy that most of the expression changes that we have identified have occurred in only one species, indicating that most differences are recent. As it becomes more feasible to carry out population surveys for expression polymorphism, it will be

important to quantify how many of these changes are fixed between species and how many are polymorphic[126,134].

The expression changes that we identified could have resulted from differences in transcript abundance (e.g., *cis*-regulatory changes) or cellular composition (e.g., expanded or contracted cell populations). Though we cannot separate these possibilities with bulk tissue samples, the fact that most changes occurred in one tissue ("dispersed") supports an evolutionary model of modular change. We suggest that the same argument applies to the cell specificity of young genes (Fig. 3C). Both observations are important because a key factor in determining anatomical evolution is the pleiotropy of mutations. Due to the functions that individual genes have across multiple tissues, it is expected that the diversification in any one tissue (or subset) will arise through mutations in the genes' modular cis-regulatory regions[135,136]. To the extent that transcript abundance drives the differences in our datasets, our results are consistent with previous findings that indicate that most between-species expression changes are driven by *cis*-regulatory modifications[133,137–140]. We expect that the

close relationships between these species will foster the identification of candidate regulatory differences that can be studied to further understand the molecular basis of transcript abundance evolution. Much less is known about the evolution of cell population sizes. In the case of *D. melanogaster*'s species-specific male-biased foreleg expression, we have found preliminary evidence that both transcript abundance and cell abundance evolution may be involved. We will soon be able to address this question more thoroughly through cross-species comparisons of single-cell atlases.

Molecular evolutionary studies of chemosensory genes have consistently highlighted their rapid protein coding and copy number evolution[141–144]. Our analyses demonstrate that changes in transcript abundance and species-specific expression gains and losses also fuel their fast evolution. It has been suggested that the cell-specific expression patterns of most chemosensory genes, along with partially overlapping molecular functions (e.g., promiscuous ligand-binding), result in relatively fewer pleiotropic constraints and, as a result, increased evolutionary freedom to change[141]. It is likely that their narrow cellular expression also allows for increased flexibility to fine-tune their levels of expression. Though the phenotypic implications of chemoreceptor expression levels remain unclear, it is plausible that they shape neuronal sensitivity or other cellular kinetics that impact a neuron's encoding of chemical information. We also have evidence from several peripheral sensory neuron populations that they can expand/contract quickly[54,145,146] and are likely contributing to species differences in chemoreceptor expression levels. More comprehensive studies are needed to assess how frequently such changes are occurring. Of potentially greater immediate phenotypic consequences are chemoreceptors' ability to gain (or lose) expression in novel tissues. We estimated that approximately a third of the chemosensory genes may have done so over the diversification of these six species. And while instances of unusual or "ectopic" receptor expression, as illustrated by *Or1a* (Fig. 4D), call for additional functional characterization, they are also a reminder of the first step that all receptors and receptor-operated channels have taken as they have diversified across tissues throughout the animal kingdom.

As with other comparative functional genomic studies, identifying the specific changes that are translated into phenotypic differences remains an outstanding challenge. The phylogenetic framework provided here will help to devise future experiments for addressing this question, as illustrated by our investigation of five chemosensory genes with species-specific expression patterns. One line of evidence pointing towards a substantial fraction of the expression differences being functionally important is our observation that they tend to be cell-specific. Though it is conceivable that a similar trend could be produced by neutral evolution (e.g., expression drift being more common among sets of genes that are cell-type-specific), we argue that this observation nonetheless provides important genome-wide evidence consistent with them being functionally relevant. This is most convincing in the context of sex differences, where nearly all expression changes are species-specific and where, in *D. melanogaster*, we linked species-specific sex-biased genes to specific cell populations involved in sexually dimorphic functions[22,128–130].

## Methods

### Fly strains, rearing, and dissections

*D. melanogaster* (NDSSC, GDL B54[147]), *D. simulans* (NDSSC,14021-0251.008), *D. sechellia* (NDSSC,14021-0271.07), *D. santomea* (NDSSC,14021-0271.00), *D. erecta* (NDSSC,14021-0224.01) and *D. suzukii* (Kyorin, K-AWA036) flies were reared on a standard yeast/cornmeal/agar medium supplemented with Carolina 4-24 Formula and maintained in a 12:12 h light:dark cycle at 25 degrees. Adults between 2 to 10 days old were sex-sorted on $CO_2$ at least 24 h before the dissections. Third instar larvae were taken directly from the food medium the day they were dissected. For each replicate, 10 third instar larval

heads, 25 proboscis, 50 legs, 5 ovipositors, and -100 antennae were collected. Three replicates were made per sex and species for the proboscis, the legs, and the antennae; 3 replicates were made per species for ovipositors and larval heads.

### Tissue collection

All adult samples were collected from flies aged between 2-10 days. Antennae were collected by flash-freezing flies in liquid nitrogen and agitating them over a mini-sieve connected to a collection dish[148]. Antennae were selected from the collection dish using a pipette under a dissecting scope. Forelegs, ovipositors, and proboscis with maxillary palps were collected from individual files using forceps and a micro scalpel under a dissecting scope. Though the proboscis and maxillary palps are distinct appendages, we combined them to reduce library and sequencing costs. Third instar larvae were collected from vials by floating them in 75% sucrose water and washed. Larva heads were removed under a dissecting scope using a micro scalpel.

### mRNA library preparation and sequencing

Dissected tissues were homogenized in 200µl of Trizol (Invitrogen) using a Precellys24 (6800 rpm, 2x30s with 10 s breaks; Bertin Technology) followed by a standard Trizol RNA extraction. The final mRNA concentration was measured using a DeNovix Ds-11 FX spectrophotometer. mRNA libraries were prepared using KAPA Stranded mRNA-seq Kit (Roche) following the manufacturer's instructions (Version 5.17). Briefly, 500 ng of total RNA diluted in 50ul of RNase-free water was first placed on supplied mRNA capture magnetic beads to allow the isolation of mature, polyadenylated mRNA, which was subsequently fragmented to a size of 100–200 bp. Double-strand cDNA was then synthesized, marked by A-tailing and barcoded with 2.5 ul of TruSeq RNA UD Indexes (Illumina). SPRI select beads (Beckman Coulter) were used for cleanup. Library concentrations were measured using Qubit dsDNA HS Assay Kits (Invitrogen). Fragment analysis and HiSeq 4000 single-end Illumina sequencing were performed by the Lausanne Genomic Technologies Facility. Information for all molecular reagents used in this project can be found in Supplementary Data 9.

### In situ hybridization chain reaction experiments

*Gene choice: Gr32a, Gr33a, Gr61a, Ir7f, Or1a,* and *Or45a* were chosen based on information on their previous functional characterization and/or expression in *D. melanogaster* (see Supplementary Fig. 17 legend) and because we found their species-specific expression differences to be the most intriguing in light of existing data.

*Probe sets:* HCR probes set, amplifiers, and buffers were purchased from Molecular Instruments. The list and the sequences of the probes used can be found in Supplementary Data 9. Coding sequences and 5' and 3'UTRs, were extracted from the species reference genomes and aligned. *D. melanogaster* sequences were used to design HCR probe sets for genes sharing >91% identity across our target species. If sequence identity was less than 91%, or if we failed to detect a signal using a *D. melanogaster* probe set in a different species where transcripts were detected in our RNA-seq dataset, we designed species-specific probe sets. Based on these criteria, species-specific probes were designed for *D. simulans Gr61a, D. suzukii Gr32a, D. sechellia Ir7f,* and *D. suzukii Gr66a.*

*In situs:* Flies between 2 to 9 days old were cold anesthetized and dissected on ice. Samples were collected on PBT (1XPBS, 0,1% Triton X-100) and fixed in 2 ml of a 4% paraformaldehyde, 1X PBS, 0.1% Triton X-100 solution at 4 °C on a rotator set at low speed (<20 rpm) during 2 h for antennae, 4 h for *D. sechellia* proboscis and 24 h for the other species' proboscis. Following fixation, samples were washed twice in PBS + 3% Triton X-100 and three times in PBT. The protocol suggested by Molecular instruments for generic samples in solution was then followed with minor adjustments (https://files.molecularinstruments.com/MI-Protocol-RNAFISH-GenericSolution-Rev9.pdf). Samples were

pre-hybridized in 300 µl of probe hybridization buffer for 30 min at 37 °C. For antenna samples, 3,5 µl of control probe (*Orco* or *nsyb*) and 5 µl of experimental probes were used. For proboscis, 5 µl of control (*nsyb*, *Gr66a*) and experimental probes were added to the amplification buffer. Samples were also pre-amplified in 300 µl of amplification buffer. For antenna samples, 6 µl of hairpin solution designed to amplify the signal of control probes was used, 10 µl otherwise. For proboscis samples, 10 µl of hairpin solutions were used to amplify both the controls and the experimental probes. After washes, samples were mounted in Vectashield and stored at 4 °C. Information for all molecular reagents used in this project can be found in Supplementary Data 9.

*Image acquisition:* Antennae, proboscis and larvae images were acquired on inverted confocal microscopes (Zeiss LSM 710 or LSM 880) equipped with an oil immersion 40X objective (Plan Neofluar 40X oil immersion DIC objective; 1.3 NA). The images were processed in Fiji (v1.53)[149].

### Gene annotations

Annotations in General Feature Format were generated for all species using BRAKER v2.1.6 and Augustus v3.4.0[150,151]. We ran BRAKER with the --etpmode flag as we provided evidence from both our aligned RNA-seq data and an orthologous protein dataset for arthropods (arthropoda_odb10). The quality of annotations was checked with BUSCO v3.0.2[152]. First, we generated fasta files with coding sequence from the annotations using Cufflinks v2.2.1[153] gffread function (`-w exons.fa -W -F -D -E -o filtered.gff` flags). Completeness was checked against the diptera_odb9 dataset. BUSCO scores were similar across species: *D. simulans* 97.3%, *D. melanogaster* 97.1%, *D. erecta* 97.0%, *D. santomea* 94.5%, *D. suzukii* 91.9%, *D. suzukii* 97.2%. The species' GTFs are in Supplementary Data 10.

### OrthoFinder-based orthology analysis

Our next goal was to group our annotated sequences into their respective orthologue groups using OrthoFinder v2.3.8[154]. The input peptide sequence was generated for each species by the following steps: (1) fasta files of coding sequence from annotations were converted to peptide sequence using the transeq function from EMBOSS v6.6.0[155], (2) duplicate genes introduced from BRAKER's pipeline were removed using a custom script (rmduplicategenes.sh), (3) Orthofinder's primarytranscript.py was run on each of the resulting peptide fasta files. These input peptide sequences were then placed in the same directory and we ran OrthoFinder to generate our orthologue groupings. We additionally added the -M msa flag to generate gene trees.

### Opposvum-based orthology analysis and gene IDs

We used Possvm[156] (v1.1) to refine orthology relationships (1 to many and many to many) inferred by Orthofinder (above). We first aligned non-1:1s orthologs using MAFFT[157] (v7.490; `mafft --auto protein.fa`) and outputted alignments in phylip format. We then generated phylogenetic trees containing bootstrap information at each node using IQ-TREE[158] (v2.2.0.5; `iqtree2 -s ./MAFFT_ortho/${spe} -mset WAG,LG -b 200`), testing for the best substitution model (WAG or LG) and performing 200 bootstrap replicates. Finally, we used Possvm to identify new orthogroups. For this step, we first parsed phylogenies using the species overlap algorithm, and second, we clustered orthogroups using the MCL clustering method. We updated the former orthofinder Orthogroup.tsv with the list of newly generated orthogroups which included 2,066 new 1:1s genes. Orthogroups were renamed according to the *D. melanogaster* reference genes, which were identified through iterative BLAST (v2.10.1+)[159]. For this step, we used tblastn to query our list of protein orthogroups on a *D. melanogaster* gene database containing nucleotide fasta from all annotated CDS. BLAST results were sorted according to their best hit (bit score selection) and the matching gene names were appended to our inferred orthogroups IDs. This "lookup table" is available as Supplementary Data 11.

### RNA-seq read mapping

Each species' Illumina reads were mapped to its own soft-masked reference genome using STAR (v2.7.8[160]), inputting the GTF files generated above. On average we obtained 43 million mapped reads per replicate, with mapping rate modes ranging between 0.79–0.90. A single *D. simulans* proboscis sample resulted in fewer mapped reads due to the amplification of a viral sequence, but was otherwise highly correlated with the two other replicates and was therefore retained. Sample replication across all tissues was high, with an overall average Pearson correlation coefficient of 0.98; the range of Pearson correlation coefficients within each tissue's replicates was between 0.97–0.99. The one exception was the ovipositor dataset, likely reflecting less precise dissections (above). Pearson correlation coefficients for the ovipositor samples ranged from 0.93–0.99, with the replicates of *D. erecta*, *D. santomea*, *D. sechellia* being more variable (average Pearson correlation coefficients = 0.93, 0.95, 0.96, respectively) than the other three species (average Pearson correlation coefficients = 0.97, 0.97, 0.99). QC files and plots can be found within the project's repository (https://gitlab.com/EvoNeuro/sensory-rnaseq), in particular, see the Full_Data_normPCA_trimmed.html and plot_mapping_stats_good_samples.html files.

### Read count table generation

*Full-length gene*: Expression count tables were generated using HTseq (v0.11.2[161]), inputting the GTF files generated above (Supplementary Data 12; the corresponding TPM table for the 1:1 orthologs is File Supplementary Data 13).

*Trimmed genes:* Despite the six species being closely related, differences in orthologous gene lengths exist. If unaccounted for, these differences may lead to misleading cross-species differential expression results when using methods that assume identical gene lengths. To account for length differences in our PCA or clustering analyses and for analysis using DESeq2 (v1.34.0[162]), we generated count tables based on orthologous gene regions that were conserved across all six species. Conserved regions were identified based on DNA alignments (MAFFT v7.475[157]) of the 1:1 orthologs. We excluded gene regions if any of the six species contained a gap greater than 150 bp (using the script get_aligned_blocks.py). Using the coordinates of the conserved gene regions, we then generated a set of "trimmed" GTF files (using the script make_trimmed_gtf.py; the species' trimmed GTFs are found in Supplementary Data 14) that were passed to HTseq (v0.11.2[161]) for computing the "trimmed" count tables (Supplementary Data 15). The "trimmed" GTF file includes the full set of genes that were annotated in each species' genome but contains the modified genic coordinates based on the conserved alignments for the set of 1:1 orthologs.

The normalized count data for 1:1 orthologs can be explored and plotted with our CT[2] dashboard available at: https://ctct.unil.ch/.

### Transcriptomic clustering and Relative rate tests

Transcriptomes were clustered by species using the set of 1:1 orthologs and a phylogenetically-informed distance measure implemented in TreeExp (v0.99.3[68]). TreeExp implements a statistical framework assuming that gene expression changes are constrained by stabilizing selection (based on the Ornstein-Uhlenbeck (OU) model). For phylogenetic reconstruction, we generated "taxa.objects" from our TPM normalized expression matrix specifying taxa (species) and sub-taxa (tissue) levels. Distance matrices were computed for each tissue by modeling gene expression changes under a stationary OU model (method= "sou"). Finally, distance matrices were converted into phylogenetic trees using the neighbor-joining method, setting *D. suzukii* as an outgroup and performing 100 bootstrap replicates.

Relative rate tests were carried out in TreeExp (v0.99.3[68]) for all pairwise comparisons using its RelaRate.test function. For these analyses, only genes with a TPM > 1 were included. To confirm that divergence score estimations were not driven by a subset of genes as well as to give stronger statistical power to the analysis, we computed divergence Z-scores by randomly sampling 1000 genes 1000 times for each species pair and each tissue sample. We compared the per species per tissue Z-score distribution from randomly sampled genes to both the minimum and maximum value of the non-significant Z-score distribution using a Wilcox test statistic in R (v4.1.2[163]).

## Differential expression for 1:1 orthologs

Evolutionary changes in gene expression were detected using the l1ou R package (v1.43[164]). l1ou uses a phylogenetic lasso method to detect past changes in the expected mean trait value, assuming traits evolve under an Ornstein–Uhlenbeck (OU) process. We used a reference species tree that was previously inferred[53] and the species' mean TPM for each gene, for each tissue, as the evaluated traits. We set the maximum number of possible expression changes to 3 (half the number of taxa in the tree) and selected the best model for the number of expression changes using the phylogenetic-informed BIC approach (pBIC). We further tested the most likely evolutionary scenario for individual gene expression trees using EvoGeneX (v0.9.9.0[165]). Using both interspecific and intraspecific (within species triplicates) information, EvoGeneX carries out tests between three models of expression evolution: (1) neutral evolution (Brownian motion), (2) constrained evolution with one expression optimum (OU model), and (3) constrained evolution with one or more gene expression optima (OU model with multiple optima values). We found that gene expression changes detected by the l1ou method strongly agreed with the most likely evolutionary scenario inferred by EvoGeneX (Supplementary Fig. 21). This analysis enabled us to quantify the number of neutrally evolving, constrained, and divergent branches for each expression tree (Fig1 E).

*Coincidental index:* For each gene, we calculated the frequency that it changed in expression in multiple tissues simultaneously by computing a simple "coincidental index" defined as:

$$\sum n_{obs(t)} / \sum n_{\max .pos(t)} \tag{1}$$

Where $n_{obs(t)}$ is the number of tissues an expression change occurred at time $t$ and $n_{\max .pos(t)}$ the maximum number of possible changes at time $t$. This index takes a value between 0 to 1 where 0 reflects no expression changes, 0.2 reflects a change that occurred in only one tissue (dispersed) and 1 an expression change that occurred simultaneously in all tissues (coincidental).

*DESeq2 analyses:* Pair-wise based identification of differentially expressed genes was carried out with DESeq2 (v1.34.0[162]) specifying the following design: ~ 1 + species + tissue + species:tissue. For these tests, the set of 1:1 orthologs (above) and the "trimmed" count tables (above) were inputted.

*Gene module analyses:* We identified co-expressed gene modules between tissues using soft-clustering algorithms implemented in CEMITools (v1.18.1[166]).

## Analyses of gene age

We performed gene age analyses on gene lists derived from ref. 167. Genes predating the speciation of the *Drosophila* subgenus (~50 My ago) were classified as "old", while new genes that have emerged since the *Drosophila* subgenus speciation event were classified as "young". Duplicated genes and their level of duplication are derived from our ortholog annotation on the set of non-1:1 orthologs (Supplementary Data 16).

## Manual curation of chemosensory gene set

Chemosensory genes were first extracted from the look-up table generated for the global dataset (Supplementary Data 16). Genes for which an ortholog was missing in one or more species, genes with multiple paralogs, or genes previously annotated in *D. melanogaster* or *D. suzukii* but missing in our datasets were investigated and manually corrected if an annotation error was identified. *D. melanogaster* coding sequences were obtained from flybase[168] and *D. suzukii* coding sequences from the literature[143]. Each species' reference genome was uploaded into Geneious (v2022.0.2) and annotated with the GTF files generated above. The coding sequences of the genes selected for manual correction were then combined with these annotated genomes using Minimap2 (v2.17[169]). A new GTF file for the chemosensory genes was generated for each species with annotation errors corrected and previously omitted missing genes added. The GTF files for these manually curated annotations are available in Supplementary Data 17).

For each tissue, the mean TPMs for each gene across replicates were calculated and the number of genes from each chemosensory family that were detected as expressed was evaluated with TPM thresholds of 0.25, 0.5, 1, 2, and 3. For the antenna and proboscis, the number of genes detected only slightly decreased with TPM thresholds between 0.5 and 2 TPM. However, for ovipositor, forelegs, and larva, the number of genes detected dropped significantly with the increase of the TPM threshold. This is likely because some genes are expressed in a few cells, leading to low TPM values. Therefore, to ensure that these genes were not excluded, the threshold for gene detection was settled at 0.5 TPM for all tissues. The TPM file for the chemosensory set of genes is available in Supplementary Data 18.

## Sex-biased gene expression

Genes that have significant differences between sexes were identified using the full set of species' genes and the "full gene" count tables. Read count data for the tissues of each species was read into DESeq2 (v1.34.0[162]) specifying the following design: ~ tissue + sex + tissue:sex. Only genes that had a normalized read count of five in three or more samples were kept for analysis. A Wald test was used to test for sex differences for each gene, requiring a log fold change of 1.5 and an adjusted $p < 0.01$ for significance.

## Fly Cell atlas data manipulation

*Data importation*: We imported 10x stringent loom and H5DA atlases of legs proboscis and antennae from flycellatlas.org[33]. The H5DA files (that contain the clustering information and feature count matrix for a subset of Highly Variable Genes) were converted to Seurat objects (Seurat v4.3.0, SeuratObject v4.1.3) using the Convert function from the SeuratDisk (v0.0.0.9020; https://mojaveazure.github.io/seurat-disk/) and were exported as RDS files using the "saveRDS" function. We used the "Connect" function from SeuratDisk to convert loom files (containing count matrix for all *D. melanogaster* genes but no clustering information) to Seurat objects and exported them as RDS files.

*Mean gene expression per cell cluster*: We split Seurat objects by cluster (subset(atlas.data, idents = "cluster_ID")) and extracted their respective feature count matrices (GetAssayData(object=atlas.data, slot = "count")). We then calculated the mean expression of individual features per cluster (rowMeans()) and log-transformed their expression for downstream analysis.

*Visualization of a gene of interest*: The AddModuleScore function from Seurat was used to select gene subsets and visualize their expression using the "FeaturePlot" function. To visualize subsets of cells expressing specific features, we used the "DimPlot" function specifying cells of interest with the "cells.highlight" option. Gene expression cutoffs were determined after visual examination to highlight highly expressing cells only.

*Cell type homology between tissue*: We used the Seurat "FindAllMarkers" (atlas.data, only.pos = TRUE, min.pct = 0.25,

`logfc.threshold = 0.25`) function from Seurat to identify significant markers ($p < 0.001$) among the top 100 list of markers per cell cluster. The list of unique shared markers was retrieved across all tissues, and we generated pairwise correlation matrices based on cluster-mean expression values for each cell cluster across each tissue. In addition, we generated pairwise matrices of the percentage of cell markers shared across tissue cell clusters. The product of these two matrices gives a score between 0 and 1, where 0 corresponds to completely unrelated cell types, and 1 corresponds to identical cell types. This homology score enabled us to cross-validate the Fly-CellAtlas annotation and to identify cell type homology across tissues at a finer scale.

### Measurements of tissue specificity and cell type specificity

We measure gene expression specificity as $\tau$[170] defined as:

$$\sum_{i=1}^{n}(1-\hat{x}_i)/(n-1); \hat{x}_i = x_i/\max(x_i) \qquad (2)$$

Where $x_i$ is the expression of the gene in tissue $i$, $n$ the number of tissues.

We apply the same formula to define $\tau$ at the level of cell clusters where $x_i$ is the mean expression of the gene in cell cluster $i$ and n the number of clusters in a given atlas. We also investigated measuring cell specificity $\tau$ index by considering $x_i$ as the percentage of cells expressing the gene in cluster $i$, which gave very similar distributions. All count values were log-transformed before applying the $\tau$ formula for stringency purposes.

### Reporting summary

Further information on research design is available in the Nature Portfolio Reporting Summary linked to this article.

## Data availability

The transcriptomic data generated in this study have been deposited in the ArrayExpress database under accession code E-MTAB-12656 and on our lab's "sensory RNAseq" GitLab repository [https://gitlab.com/EvoNeuro/sensory-rnaseq]. The publicly available single-nucleus datasets used in this study were part of the Fly Cell Atlas and are available at https://flycellatlas.org. The location of the source data for Figs. 1–5 and Figures Supplementary 1–3, 7–13 are provided in the table "Source_data.xlsx". Source data are provided with this paper.

## Code availability

Code used for this project is available on our lab's "sensory RNAseq" GitLab repository: https://gitlab.com/EvoNeuro/sensory-rnaseq.

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

## Acknowledgements

We thank Margarida Cardoso-Moreira for discussions and advice throughout the project. Margarida Cardoso-Moreira, John Pannell, Thomas Auer, Giulia Zancolli, and Lucia L. Prieto Godino provided important comments on an earlier version of the paper. T. M. Balls

provided support and inspiration. Wolf Huetteroth provided anatomical assistance with *Or1a* expression in the proboscis. Sequencing was performed at the Lausanne Genomic Technologies Facility, and the University of Lausanne's HPC services provided resources and support. Microscopy was performed at the University of Lausanne's Cellular Imaging Facility. Research in JRA's laboratory was supported by the University of Lausanne and the Swiss National Science Foundation (grants PP00P3_176956 and 310030_201188).

## Author contributions

Conceptualization: G.B., B.S.L., and J.R.A.

## Competing interests

The authors declare no competing interests.
