## [Peer Review File · Nature Communications]

Evolution of chemosensory tissues and cells across ecologically diverse DrosophilidsReviewers' Comments:

Reviewer #1:

Remarks to the Author:

How existing traits adapt and novel phenotypes arise are some of the most central questions of evolutionary biology. It is well accepted that changes in the regulation of genes are a major driver of organismal diversification, however our understanding of how gene expression evolves broadly across taxa remains unclear, especially on higher resolution scales such as cell types and tissues. In order to map out the evolution of gene expression, Bontou & Saint-Leandre and colleagues sequenced four chemosensory tissues in two developmental stages of two sexes across a clade of six closely related *Drosophila* species, producing an impressive, high-quality dataset unmatched in its depth and breadth. The authors use this data to carefully and meticulously describe the properties of gene expression evolution across species, sexes, and tissues, utilizing the power of their own data in combination with existing resources such as the *D. melanogaster* Fly Cell Atlas. This way they not only provide a most valuable addition to the *Drosophila* genetic resources, but more importantly can characterize patterns of gene expression diversification across chemosensory tissues throughout a large clade of species. They go beyond the pure description of in silico derived expression differentiation but test these on candidate genes by fluorescence in situ hybridization across the entire clade. Due to their robust data generation, integrative approach, and careful analysis the authors show how chemosensory tissues evolve across species and sexes, revealing modes of diversification between different tissues at an impressive level of detail.

This is a highly relevant contribution to our understanding of how chemosensory tissues diversify, with importance to evolutionary biology but extending to other fields including neuroscience and ethology. Overall, the manuscript is well written, the data generation followed rigorous standards, the analyses are sound, as are the conclusions drawn from it. I do not have any major concerns with any aspect of this work, it could be published as is. That said, the manuscript could gain some additional richness if a few more analyses were considered.

1. The analyses presented mainly focus on 1:1 orthologous genes. While this makes sense given that the authors are interested in understanding the evolution of gene expression across all species, it seems like a missed opportunity not to look at the expression evolution of paralogous genes that arose after the most recent common ancestor of the 6 species as well as genes that have been lost in some taxa as well. Gene birth and death is a major driver of chemosensory systems evolution across animals and thus an understanding of how duplicated genes evolve is highly relevant. The authors touch on this subject by looking at gene expression of new genes. These, however, are defined as having arisen <50MYA, which predates the MRCA of the species (15MY of divergence time). The authors could make great use of their unique dataset and look more explicitly at the patterns of expression following gene duplication. Are they expressed in the same or different tissues as their paralogous genes? How do cis-regulatory elements evolve post duplication? Similarly, what types of genes get lost through pseudogenization? Are there any discernible patterns?

2. It would be interesting to test whether the diversification of the coding and/or non-coding sequence is a predictor of gene expression evolution for chemosensory tissues. Further, contrasting an analysis of coding gene evolution with the evolution of cis-regulatory elements with respect to expression diversification could reveal important insights into how chemosensory systems evolve. Functional evolution is a major focus in the sensory neuroethology/evolution fields and such analyses have the potential for deeper insights and hypothesis generation.

Further minor comments:

-The word breadth is often mistakenly written as 'breath' throughout the manuscript.

-The closer the cell breadth of a gene is to 1, the more cell-type specific it seems to be. This is counter

intuitive. The reason to as to why could be introduced more explicitly or the metric could be changed.

-The last Results paragraph (Pages 12 and 13) is a bit confusing and could be clarified further. It is not easily accessible what comparison the patterns described allude to, especially towards the end of the paragraph.

Reviewer #2:

Remarks to the Author:

Bontonou et al. generated transcriptomic data for five chemosensory tissues obtained from both males and females of six *Drosophila* species. The authors employ computational strategies to evaluate evolutionary changes in gene expression between chemosensory tissues in distinct *Drosophila* species. They integrate their bulk RNA-sequencing findings with a published cell-atlas dataset and utilize multiplex in situ hybridization across multiple *Drosophila* species, both of which bolster their computational and sequencing approaches. Importantly, the authors created a user-friendly dashboard for other researchers to explore and utilize their datasets. Overall, this manuscript is well-written, the data is nicely depicted, and it should be a useful resource for the community.

Below are some critiques that the authors should address in their revision.

Major concerns

1) In the abstract the authors state that their analyses revealed strong evidence that expression changes of DEGs is driven by a combination of cis-regulatory and cell composition evolution. However, I do not see strong evidence that the authors directly evaluated either of these. I suggest removing or softening this language.

2) There are several places in the text where the authors superficially describe that they identified changes in gene expression in distinct tissues in different species, for instance in the text relating to figures 2 and 3. It would be useful and interesting to know what kinds of genes are changed. Are they those that affect gene expression of other genes? Are they related to chemosensation?

3) Ideally the proboscis and palp tissues would have been sequenced separately given that these organs detect mostly distinct chemosensory cues. The authors should at least provide a compelling reason/discussion for sequencing the proboscis and maxillary palps together. Relatedly, it should be clarified whether the authors believe that the proboscis genes are primarily driving the clustering of the proboscis + palp transcriptomes in Fig 1B. If proboscis and palps were sequenced separately, do the authors expect the palps to cluster closer to the antenna?

4) The authors contrast their PCA results (tighter clustering of forelegs and proboscis + palp samples than proboscis + palp and antenna samples) with a study detailing the role of Hox genes in specification of *D. melanogaster* maxillary palps and other chemosensory organs. The comparison between transcriptomes of adult tissues and the expression of a handful of tissue patterning genes is confusing. Do the authors expect tissues that are more similar in development to maintain this similarity into adulthood? Or is there another reason why this is highlighted in the text?

5) The text describing Fig. 3 is a bit convoluted. It might be helpful to have a schematic describing the hypothesis and results to aid the reader.

6) Please clarify the criteria and rationale for choosing the specific genes depicted in Fig. 3 B-C.

7) Based on the density plot in Fig. 3C (top) Or43b expression is not entirely restricted to antennae; however, there does not seem to be significant expression (beyond a few dispersed cells) of this gene

in the plots from the FCA dataset. Is there a discrepancy between the author's data and the published snRNA-seq data? Also, what is the rationale for displaying the specific in Fig. 3?

8) Please clarify how the genes for the HCR experiments were chosen. What happened to Gr32a? Though the text says there is co-expression for Gr32a and Gr33a, I cannot find the Gr32a data.

9) In Fig. 4C, is Gr61a mRNA expressed in the sacculus or is this nonspecific labeling?

10) The text related to Fig. 5 represents some of the most significant discussion of specific genes that display differences between experimental groups. However, the plots in Fig. 5 E-F do not display any of the genes discussed and it is unclear if those plots are from male-specific FCA data or if they contain nuclei from both sexes. It would be helpful have example plots of the genes that are different in each sex in each of the cell-types discussed.

Minor concerns

1) Page 9, line 26: typo: 'specie's'

2) Please make clear that Fig. 2C is a model.

3) High-resolution, zoomed in images of the cells expressing chemosensory receptors in each organ in Fig. 4 and co-expression of cell markers and receptors in Fig. S13 are needed. For example, it is challenging to evaluate co-expression in Fig. S13D.

4) Y-axis labels are missing from all plots in Fig. 5B-C.

Reviewer #3:

Remarks to the Author:

This publication by Bontonou et al. presents an evolutionary comparison of chemosensory tissues gene expression across five species of *Drosophila*. This is an important dataset that adds to the rich understanding of evolution of this important system. The major findings of the work are that gene expression evolution is dominated by stabilizing selection and that most chemosensory genes have undergone changes in gene expression level. They also find that some species have sex-specific changes in gene expression and link these changes to particular cell types. This study is a valuable addition to the field and would be strengthened by solidifying the major conclusions and more deeply analyzing their results.

Major comments:

1. One of the main conclusions of the paper is that stabilizing selection is the major evolutionary force acting on transcriptomes of these chemosensory tissues. The evidence for this seems limited to the relative rate test that examines relative transcriptome divergence of pairs of species, calibrated by an outgroup. Although it is expected that stabilizing selection would be dominant, if this is a major conclusion, it should be strengthened. Does the choice of outgroup species affect the which species show divergence? Quantifying the total number of genes that are conserved in the analysis in Fig. 2 would also lend support to the stabilizing selection argument. It would also be a more significant to compare the overall divergence of chemosensory tissues versus the divergence of tissues expected to show greater conservation (using previously published datasets).

2. It would strengthen the paper to validate the results by comparing to previously published datasets. These studies are less comprehensive than the present study but would provide strong evidence for the quality of the present dataset.

3. One significant conclusion of the paper is that both cell type changes and expression within cell types contribute to gene expression evolution. However, the section where this is examined (p. 13 line 446) is referred to as preliminary. This conclusion should be strengthened if it is a major conclusion. One prediction is that if cell types are changing number, then most or all of the genes in the cell type would change expression in the same direction. If gene expression is changing within a cell type then

the most coherent expression changes would likely be seen at the pathway level (examined in Fig. S3). Testing these predictions and differentiating between these modes of evolution would greatly strengthen the paper.

4. A downside of this study is the lack of connection between gene expression evolution in these chemosensory tissues and functional consequences. Figure 4 shows interesting shifts in chemosensory gene expression, but not any possible functions of these changes. This could be remedied by experiments to test the consequences or validation of previous studies with the present dataset. I.e. past studies that have found functional consequences of gene expression changes are captured by this dataset.

Minor questions and comments:

1. The findings in Fig 1C and 1D for the ovipositor data seem to conflict. If *D. santomea* and *D. erecta* diverge at lower rates, why do they not cluster together more closely? Does this call into question the value of the analysis in 1C?
2. Do the de novo gene annotations depend on genome quality?
3. The paragraphs p. 5 line 196 and line 204 leave it unclear whether for these analyses divergence of a gene was considered in multiple or single species. This information should be specified in Figure 2 as well. It would be helpful to give some
4. The analysis in Fig S5A seems to suggest that coincidental expression changes are inversely correlated with branch length. The authors should indicate whether they predict that these are resolved by later compensatory mutations to resolve pleiotropic mutations. This could be examined by inferring coincidental changes in internal branches that were later resolved by compensatory mutations in a subset of later branches.
5. P. 7 line 243 – “Our finding that most differentially expressed genes are broadly expressed” – it was not clear to me where this was already shown at this point in the paper. This statement seems contradicted by the statement in the next sentence that “genes that have changed in expression have similar modes of tissue breadth but tend to be more tissue-restricted than genes that have not changed.” More clarity would be helpful here.
6. The analysis in Figure 3 is a strength of the paper. In several places (e.g. p.7 line 246, 254) “breadth” should be “breadth”. I also suggest switching tissue breadth to tissue specificity because the scale goes from 0 to 1 where 1 is the least breadth or greatest specificity. Alternatively, you could swap 0 and 1 to better reflect the measurement of breadth.
7. The publication mentions that all of the genes chosen for functional validation confirmed the RNA-seq results except for Gr32a. How did Gr32a differ from the RNA-seq results?

RESPONSE TO REVIEWER COMMENTS

Reviewer #1 (Remarks to the Author):

How existing traits adapt and novel phenotypes arise are some of the most central questions of evolutionary biology. It is well accepted that changes in the regulation of genes are a major driver of organismal diversification, however our understanding of how gene expression evolves broadly across taxa remains unclear, especially on higher resolution scales such as cell types and tissues. In order to map out the evolution of gene expression, Bontou & Saint-Leandre and colleagues sequenced four chemosensory tissues in two developmental stages of two sexes across a clade of six closely related *Drosophila* species, producing an impressive, high-quality dataset unmatched in its depth and breadth. The authors use this data to carefully and meticulously describe the properties of gene expression evolution across species, sexes, and tissues, utilizing the power of their own data in combination with existing resources such as the *D. melanogaster* Fly Cell Atlas. This way they not only provide a most valuable addition to the *Drosophila* genetic resources, but more importantly can characterize patterns of gene expression diversification across chemosensory tissues throughout a large clade of species. They go beyond the pure description of in silico derived expression differentiation but test these on candidate genes by fluorescence in situ hybridization across the entire clade. Due to their robust data generation, integrative approach, and careful analysis the authors show how chemosensory tissues evolve across species and sexes, revealing modes of diversification between different tissues at an impressive level of detail.

This is a highly relevant contribution to our understanding of how chemosensory tissues diversify, with importance to evolutionary biology but extending to other fields including neuroscience and ethology. Overall, the manuscript is well written, the data generation followed rigorous standards, the analyses are sound, as are the conclusions drawn from it. I do not have any major concerns with any aspect of this work, it could be published as is. That said, the manuscript could gain some additional richness if a few more analyses were considered.

We thank this reviewer for their enthusiastic assessment and thoughtful suggestions.

1. The analyses presented mainly focus on 1:1 orthologous genes. While this makes sense given that the authors are interested in understanding the evolution of gene expression across all species, it seems like a missed opportunity not to look at the expression evolution of paralogous genes that arose after the most recent common ancestor of the 6 species as well as genes that have been lost in some taxa as well. Gene birth and death is a major driver of chemosensory systems evolution across animals and thus an understanding of how duplicated genes evolve is highly relevant. The authors touch on this subject by looking at gene expression of new genes. These, however, are defined as having arisen <50MYA, which predates the MRCA of the species (15MY of divergence time). The authors could make great use of their unique dataset and look more explicitly at the patterns of expression following gene duplication. Are they expressed in the same or different tissues as their paralogous genes? How do cis-regulatory elements evolve post duplication? Similarly, what types of genes get lost through pseudogenization? Are there any discernible patterns?

The reviewer makes a good point that the duplication/deletion of chemosensory genes is an important aspect of their evolution and that our analyses would be improved by expanding beyond 1:1 orthologs. We have added a new analysis that summarizes the expression changes of lineage-specific chemosensory duplications. The new supplementary figures 14 - 16 display the results of these analyses and we have added a new supplementary table (Supp. Table 7), with the underlying data. We have added the following text to the manuscript on pg. 10:

“Similar analyses of the set of 85 chemosensory genes (from 31 gene families) that have duplicated since the common ancestor of the six species revealed that nearly all recent paralogs have retained expression in the same tissues but often at lower levels (Fig. S14-15, Table S7). Two exceptions are *Gr59a4* and *Ir52f2* in *D. suzukii*, which show gains of increased expression levels that may indicate the neofunctionalization of these genes (Fig. S16).”

2. It would be interesting to test whether the diversification of the coding and/or non-coding sequence is a predictor of gene expression evolution for chemosensory tissues. Further, contrasting an analysis of coding gene evolution with the evolution of cis-regulatory elements with respect to expression diversification could reveal important insights into how chemosensory systems evolve. Functional evolution is a major focus in the sensory neuroethology/evolution fields and such analyses have the potential for deeper insights and hypothesis generation.

We are indeed very interested in the question of cis-regulatory changes and their relation to the expression changes that we have found. We are currently carrying out analyses for an independent project that uses chromatin accessibility data (bulk and single-cell ATAC-seq) that involves the examination of genome alignments and scans for indels/TE insertions in putative and annotated regulatory regions. These analyses will form the basis for experimentally verifying their role in the changes we have quantified in this manuscript.

Regarding the relationship between coding and expression evolution, many past analyses have examined this relationship in flies and other taxa. The results have been quite mixed depending on a number of factors (e.g., Drummond et al. PNAS, 2005; Lemos et al. MBE, 2005; Kohn et al. Genes and Genetic Systems, 2008; Warnerfors and Kaessmann, GBE, 2013; Biswas, Kakali, et al. Genomics 2016; Guillén et al. J. of Heredity, 2019).

Though relating coding evolution to expression evolution was not an aim of this study, we have now investigated if rates of amino acid change are correlated with levels of gene expression variation between species. To do this we quantified protein evolution, as measured by averaged dN/dS along the branches of the *D. melanogaster* subgroup (as summarized from Stanley and Kluantinal et al. 2016) and tested if they were correlated with expression variance (from *D. melanogaster*, *D. simulans*, *D. sechellia* and *D. erecta*). We found a positive correlation between dN/dS of the *D. melanogaster* subgroup species and between-species gene expression variance for *D. melanogaster*, *D. simulans*, *D. sechellia* and *D. erecta* (Fig. 1), indicating that proteins evolving at faster rates also exhibit higher expression differences between species. However, this broad correlation is not a direct predictor of gene expression changes. As has been shown before, other variables impact this relationship including level of gene expression (Pál et al. Genetics, 2001; Krylov, Dmitri M., et al. Genome Research 2003; Rocha, Eduardo PC, and Antoine Danchin, MBE, 2004), lineage-specific demographic effects (e.g., Hutter, Stephan, et al. Genome Biology, 2008; Zhao, Li, et al. PLoS Genetics, 2015; Huang, Yuheng, et al. Genetics, 2021; Storey, John D., et al. Am. Journal of Human Genetics, 2007) and changes in upstream regulatory sequences (as mentioned above). Currently, we do not feel that this analysis brings much to the current manuscript and may instead be a distraction. We think this line of inquiry is important, but we prefer to include these analyses in the independent project we are working on where we are looking at the genetic bases of the expression changes using chromatin accessibility data.

Fig. 1. Relationship between mean dN/dS

dN/dS estimates were taken from FlyDivas (<http://www.flydivas.info/>) and expression variation was calculated as the coefficient of variation of gene expression between species. Species considered for the analysis are *D. melanogaster*, *D. simulans*, *D. sechellia* and *D. erecta*. Dashed lines show Pearson's correlation coefficient for each tissue.

Further minor comments:

-The word breadth is often mistakenly written as 'breath' throughout the manuscript.

Thanks for catching this. Due to reviewer's suggestions (below) all use of "breadth" has been substituted for "specificity".

-The closer the cell breadth of a gene is to 1, the more cell-type specific it seems to be. This is counter intuitive. The reason to as to why could be introduced more explicitly or the metric could be changed.

We note that reviewer 3 had the same issue. We have now substituted "breadth" for "specificity" in the y-axis of Figure 3B/C and Figure 5D and their associated legends. And for consistency, we did the same throughout the Results and Methods sections.

-The last Results paragraph (Pages 12 and 13) is a bit confusing and could be clarified further. It is not easily accessible what comparison the patterns described allude to, especially towards the end of the paragraph.

We have reworked wording in this paragraph to make it clearer (which also involved improving some of the analyses in line with reviewer 3's comments - point #3, below), and also split it into two paragraphs.

Reviewer #2 (Remarks to the Author):

Bontonou et al. generated transcriptomic data for five chemosensory tissues obtained from both males and females of six *Drosophila* species. The authors employ computational strategies to

evaluate evolutionary changes in gene expression between chemosensory tissues in distinct *Drosophila* species. They integrate their bulk RNA-sequencing findings with a published cell-atlas dataset and utilize multiplex in situ hybridization across multiple *Drosophila* species, both of which bolster their computational and sequencing approaches. Importantly, the authors created a user-friendly dashboard for other researchers to explore and utilize their datasets. Overall, this manuscript is well-written, the data is nicely depicted, and it should be a useful resource for the community.

We thank the reviewer for their positive assessments and careful reading.

Below are some critiques that the authors should address in their revision.

Major concerns

1) In the abstract the authors state that their analyses revealed strong evidence that expression changes of DEGs is driven by a combination of cis-regulatory and cell composition evolution. However, I do not see strong evidence that the authors directly evaluated either of these. I suggest removing or softening this language.

This is a fair point. We have removed the sentence from the abstract.

2) There are several places in the text where the authors superficially describe that they identified changes in gene expression in distinct tissues in different species, for instance in the text relating to figures 2 and 3. It would be useful and interesting to know what kinds of genes are changed. Are they those that affect gene expression of other genes? Are they related to chemosensation?

We agree that in addition to the global quantification of the changes, it is important for us to categorize the types of genes that changed. Because there are many genes that evolved expression changes, we provide a graphical summary of their functional enrichments in Fig. S3. We now also provide a new table that underlies the enrichments in Fig. S3 so that genes and categories can be easily searched (Table S2). Hopefully this makes it easier to observe that genes with changes in gene expression include chemosensory-related genes as well as genes with many other functions including multiple transcription factors.

3) Ideally the proboscis and palp tissues would have been sequenced separately given that these organs detect mostly distinct chemosensory cues. The authors should at least provide a compelling reason/discussion for sequencing the proboscis and maxillary palps together. Relatedly, it should be clarified whether the authors believe that the proboscis genes are primarily driving the clustering of the proboscis + palp transcriptomes in Fig 1B. If proboscis and palps were sequenced separately, do the authors expect the palps to cluster closer to the antenna?

We see the reviewer's point: the maxillary palps are primarily olfactory organs that develop from the eye-antennal imaginal discs while the proboscis develops from the labial discs. We agree that it would have been ideal to separate the maxillary palps from the rest of the proboscis, however we the considerable extra costs were a limiting factor. And though we could have removed the maxillary palps prior to extracting mRNA, this would have prevented us from examining potential species-differences in Olfactory receptors expressed in the samples. We now note this decision explicitly the Methods (under Tissue Collection). We now also note that the overlap between the legs and proboscis+palps is "likely driven primarily by the relative abundance of the proboscis tissue." We suspect that dissected maxillary palp samples would cluster closer to the antenna because both tissues contain olfactory sensory neuron populations. The sets of Odorant receptors expressed between the two are quite distinct, however, as seen in Sup. Fig. 8 and as previously described

(e.g., Couto et al., 2005; Vosshall and Stocker, 2007; Dweck et al., 2016). As a result, the proximity of the clustering might depend on the diversity of other cell types found in the two appendages.

4)The authors contrast their PCA results (tighter clustering of forelegs and proboscis + palp samples than proboscis + palp and antenna samples) with a study detailing the role of Hox genes in specification of *D. melanogaster* maxillary palps and other chemosensory organs. The comparison between transcriptomes of adult tissues and the expression of a handful of tissue patterning genes is confusing. Do the authors expect tissues that are more similar in development to maintain this similarity into adulthood? Or is there another reason why this is highlighted in the text?

We agree with the reviewer that these sentences were unclear and added confusion. We have removed them.

5)The text describing Fig. 3 is a bit convoluted. It might be helpful to have a schematic describing the hypothesis and results to aid the reader.

We have reworked the text related to Fig. 3 as well as its legend to improve clarity. We have also added to our discussion to make our interpretation of these observations clearer:

“The expression changes that we identified could have resulted from differences in transcript abundance (e.g., cis-regulatory changes) or cellular composition (e.g., expanded or contracted cell populations). Though we cannot separate these possibilities with bulk tissue samples, the fact that most changes occurred in one tissue (“dispersed”) supports an evolutionary model of modular change. We suggest that the same argument applies to the cell specificity of young genes (Fig. 3C). Both observations are important because a key factor in determining anatomical evolution is the pleiotropy of mutations. Due to the functions that individual genes have across multiple tissues, it is expected that the diversification in any one tissue (or subset) will arise through mutations in the genes’ modular cis-regulatory regions.”

6)Please clarify the criteria and rationale for choosing the specific genes depicted in Fig. 3 B–C.

We select genes because they were functionally characterized genes (not CGs) that were also useful for illustrating the way that a gene’s expression specificity can vary between bulk and single-cell datasets. To clarify this, we have updated the figure legends for panels B and C to now include the following sentence:

“Colored circles with lines above the density plots indicate the expression specificity values of three previously characterized genes that were chosen to illustrate varying relationships between expression specificity at the levels of tissues and at the levels of cells (see text).”

7)Based on the density plot in Fig. 3C (top) *Or43b* expression is not entirely restricted to antennae; however, there does not seem to be significant expression (beyond a few dispersed cells) of this gene in the plots from the FCA dataset. Is there a discrepancy between the author’s data and the published snRNA-seq data? Also, what is the rationale for displaying the specific in Fig. 3?

We appreciate this detailed check. In our bulk RNA-seq data *Or43b* expression in the antenna is ~100 TPM and is ~5 TPM in the other tissues. Given the low expression of *Or43b* in the non-antenna tissues, and the well-known problem of “dropouts” for low/moderate gene expression in single-cell data, is it not surprising that only a few cells show expression for this gene outside of the

antenna. We also note that we log-transformed expression values before computing the tissue specificity index for stringency purposes and this makes *Or43b* less tissue-specific than when calculating the same index from non-log-transformed gene expression values.

As above, the three genes were functionally characterized and useful for illustrating the way that a gene's expression specificity can vary between bulk and single-cell datasets. We also added the same note to the legend for this panel.

8) Please clarify how the genes for the HCR experiments were chosen. What happened to *Gr32a*? Though the text says there is co-expression for *Gr32a* and *Gr33a*, I cannot find the *Gr32a* data.

The genes for the HCR experiments were chosen because their chemosensory functions had been previously studied and/or their expression had been described in *D. melanogaster* (please see Fig. S17's legend for the details) and because we found their novel species-specific expression differences to be the most intriguing in light of the aforementioned data. We have now added these details to the Methods section (under *in situ* hybridization chain reaction experiments).

Regarding the statement of "co-expression", our language was ambiguous. We had intended it to mean co-expression between our genes of interest and the marker of olfactory sensory neurons (*Orco*) or the pan-neuronal marker (*nsyb*). We have rewritten these sentences to be clearer. The of *Gr32a* and *Gr33a* co-expression was additionally a typo and should have read *Gr61a* and *Gr33a*. This has been corrected. We have also expanded on our negative results regarding *Gr32a*:

"To gain spatial and cellular resolution for the expression of a subset of 95 chemosensory genes with novel expression patterns, we designed *in situ* hybridization chain reaction (HCR) experiments for six of them: *Gr32a*, *Gr33a*, *Gr61a*, *Ir7f*, *Or1a*, *Or45a* (Methods). We detected expression that was consistent with our RNA-seq results for all of these genes except *Gr32a* (Figs. 4B-F, Figs. S17). For unknown reasons we were unable to detect *Gr32a* in *D. suzukii* antenna despite detecting the expected expression in the labial palps (Fig. S18)."

The new Fig. S18 displays the results of the RNA-seq count data and *Gr32a* HCR experiments in *D. suzukii*'s proboscis and antenna.

We are uncertain why we were unable to detect *Gr32a* in *D. suzukii* antenna. As we could detect *Gr32a* expression in the proboscis we know that the probe works. One explanation might be that the tissues in the antenna of *D. suzukii* prevented sufficient penetration of the probes. We encountered this situation with several of *D. sechellia* tissues, which required specific optimization of our protocol (noted in the Methods section). While we unsuccessfully tried similar modifications to the protocol for *D. suzukii* antenna, perhaps there are successful parameter combinations that we have not tried.

9) In Fig. 4C, is *Gr61a* mRNA expressed in the sacculus or is this nonspecific labeling?

It is not uncommon for the sacculus to appear slightly illuminated due to the shape of the cuticle around it resulting autofluorescence. We have added this note to the legend of Fig. 4.

10) The text related to Fig. 5 represents some of the most significant discussion of specific genes that display differences between experimental groups. However, the plots in Fig. 5 E-F do not display any of the genes discussed and it is unclear if those plots are from male-specific FCA data or if they contain nuclei from both sexes. It would be helpful have example plots of the genes that are different in each sex in each of the cell-types discussed.

We thank the reviewer for this comment and question. We note that we did not use the FCA to quantify sex differences because it cannot accurately quantify lowly/moderately expressed genes and has only 2 samples per sex. Instead, we used the FCA to identify which cell populations the sex-bias genes (as identified with the bulk RNA-seq experiments) localized to. For this reason, it makes sense to profile sex-biased gene expression on a cell atlas containing a mix of both male and female cells. In addition to this, however, we have taken the reviewers good suggestion to individually display the genes summarized in (what is now) Fig. S20. These have been added as panels C and D.

1)Page 9, line 26: typo: 'specie's'

Thank you for catching this.

2)Please make clear that Fig. 2C is a model.

We have updated the legend for Figure 2C to read:

“Schematic illustrating a hypothetical gene having expression changes involving multiple tissues that were coincidental (occurring on a single branch) or dispersed (occurring across multiple branches).”

3)High-resolution, zoomed in images of the cells expressing chemosensory receptors in each organ in Fig. 4 and co-expression of cell markers and receptors in Fig. S13 are needed. For example, it is challenging to evaluate co-expression in Fig. S13D.

Thanks for this suggestion. We have updated this supplemental figure (now Fig. S17) with high resolution images of proboscis stainings and “zoomed in” insets of that provide improved visualization of co-expression.

4)Y-axis labels are missing from all plots in Fig. 5B–C.

We have considered adding Y-axis labels but it is not conventional to have these for upset plots. The row labels in combination with the “pairings” that are displayed by the connected dots vary across columns (which are summarized by the bar plot). We prefer to leave these upset plots as they are.

Reviewer #3 (Remarks to the Author):

This publication by Bontonou et al. presents an evolutionary comparison of chemosensory tissues gene expression across five species of *Drosophila*. This is an important dataset that adds to the rich understanding of evolution of this important system. The major findings of the work are that gene expression evolution is dominated by stabilizing selection and that most chemosensory genes have undergone changes in gene expression level. They also find that some species have sex-specific changes in gene expression and link these changes to particular cell types. This study is a valuable addition to the field and would be strengthened by solidifying the major conclusions and more deeply analyzing their results.

We very much appreciate these positive comments and helpful suggestions.

Major comments:

1. One of the main conclusions of the paper is that stabilizing selection is the major evolutionary force acting on transcriptomes of these chemosensory tissues. The evidence for this seems limited to the relative rate test that examines relative transcriptome divergence of pairs of species, calibrated by an outgroup. Although it is expected that stabilizing selection would be dominant, if this is a major conclusion, it should be strengthened. Does the choice of outgroup species affect the which species show divergence? Quantifying the total number of genes that are conserved in the analysis in Fig. 2 would also lend support to the stabilizing selection argument. It would also be a more significant to compare the overall divergence of chemosensory tissues versus the divergence of tissues expected to show greater conservation (using previously published datasets).

The reviewer raises the good point that it's possible that the choice of the outgroup species affects the pairwise comparisons. This would especially be the case if the expression matrix from the outgroup species shared more similarities with one (or more) species compared to others.

We first want to explain our original decision to use *D. suzukii* as the sole outgroup species. According to the 'transcriptome-based species tree' (Figure 1C), *D. suzukii* is always identified as the outgroup. This is in contrast to *D. erecta* and *D. santomea* for which transcriptome similarities to other species vary depending on the tissues (e.g., larva head compared to the ovipositor). Given the robust placement of *D. suzukii* as both the genetic/true and "transcriptional" outgroup, we reasoned that using it as the single outgroup shared across all of the tests was justified.

But the reviewer good point still stands, and thus in order to test if using other outgroup species impacts our conclusions, we carried out analogous analyses using *D. santomea* and *D. erecta* as outgroup species to *D. melanogaster*, *D. sechellia*, and *D. simulans*. We repeated the relative rate tests and then asked if the resulting distributions of Z-scores differed from those when *D. suzukii* was used. We have included a summary figure of these analyses as a new supplementary figure (Fig. S3). We found that the new results agreed with those displayed in Figure 1D: The distributions of the Z-score are predominantly not different from zero regardless of the outgroup used, consistent with the transcriptomes having evolved primarily under stabilizing selection. In the few instances where we identified accelerated rates of divergence using *D. suzukii* as the outgroup (*D. simulans* female antenna and larva samples, *D. melanogaster's* male legs and ovipositor samples), the same species/tissues were identified when using *D. santomea* or *D. erecta* as outgroups. The single exception was the analysis involving *D. simulans' ovipositor* samples, which did not result in the elevated rates of divergence in the new analyses compared to those shown in Figure 1D. The reason for this discrepancy, however, is precisely due to the exclusion of *D. erecta* and *D. santomea* from the analyses (and thus from the loss of their high Z-scores from the overall distribution). In our updated Fig. 1D, the distribution of Z-scores for *D. simulans' ovipositor* samples is bimodal. The mode slightly below zero corresponds to the *D. simulans - D. sechellia* and *D. simulans - D. melanogaster* distances and the second mode above zero corresponds to *D. simulans - D. santomea* and *D. simulans - D. erecta* distances. As a result of using *D. suzukii* as the outgroup to the five other species, Figure 1D captures the elevated expression differences driven by the *D. simulans* comparisons to *D. erecta* and *D. santomea*. The exclusion of *D. erecta* and *D. santomea* from the new analyses similarly explains the lower divergence rates (in Fig. S3) for the antenna of female *D. melanogaster*, the male forelegs of *D. simulans*, and *D. sechellia's* larva and ovipositor samples when compared to the results shown in Figure 1D.

While addressing this comment, we found a minor error due to a species labeling bug: *D. melanogaster* and *D. simulans* names were swapped for the ovipositor analyses in Figure 1D. This

error did not impact/change any conclusions or significant results but it did reveal the bimodal distribution of Z-scores that we discuss above.

We really appreciated the suggestions for this additional work on the relative rate tests. They have added strength to our analyses and have been constructive in our thinking about the results. Given the consistency in the results across outgroups, we believe that *D. suzukii* remains the optimal choice for the analyses summarized in the Figure 1D, but have added the additional note on pg. 5 and the new Fig. S3:

“We obtained consistent results when examining the distribution of Z-scores based on subsampled sets of the 1:1 orthologs (Fig. 1D) and when using either *D. erecta* or *D. santomea* as outgroup species to *D. simulans*, *D. sechellia*, and *D. melanogaster* (Fig. S3).”

We agree with the reviewer that additional support for our claims of selective constraint would strengthen our manuscript. The low rates of between-species divergence as evidenced by the relative rate tests are consistent with similar rates of neutral evolution over the relatively short timespans or stabilizing selection (or both). We also agree that further quantification of the number of genes found to be under constraint (or not) is insightful.

We have added additional analyses to address these questions/concerns. First, we carried out a new phylogenetically-informed analysis that explicitly tested whether the between-species expression data for each gene is best explained by a neutral (Brownian motion) or a constrained (an Ornstein-Uhlenbeck with one or more optima) model. This new analysis is similar to the one I1ou-based analyses underlying the results in Fig. 2A, but while the latter is capable of identifying the branches where the expression changes occurred, the new analysis (using EvoGeneX) is capable of quantifying the proportion of neutral versus constrained genes. The overlap between the methods for identifying *which* genes have evolved expression changes is very high - between 92% and 96%, depending on the tissue, and we have created a new figure (Fig S22) showing this. Overall, we found that between 52% - 57% of genes are significantly better explained by a constrained/OU model (new Supplementary Fig. S20). Among genes that fit the constrained/OU models best, a minority were constrained by a single expression optimum (4-12%) and instead most genes were found to have at least two expression optima (88-96%, which is also consistent with the results of the I1ou-based analyses; see Fig. S20). As a result, we additionally quantified the proportion of branches - across all gene trees - that were inferred to have evolved neutrally (Brownian model), or under stabilizing selection (OU models with one or more optimum). We inferred that a slight majority of edges are under stabilizing selection (52%), followed by neutral edges (40%) and “adaptive” edges (8%; see updated Fig. 1’s panel E). We note that though EvoGeneX refers to this latter model as “adaptive” we prefer to interpret these patterns of expression change more cautiously as “divergent”. Overall, then, we estimate that ~60% of total branches for the set of 1:1 orthologs have evolved non-neutrally in these chemosensory transcriptomes.

We very much thank the reviewer for the valuable comments on these evolutionary analyses. This additional work has certainly strengthened the manuscript. In light of the reviewer’s recommendations and our new results, we have softened the “stabilizing selection language” related to Fig. 1 and only begin discussing evolutionary constraints later in the paper following the above analyses. We now also place our estimates on the number of genes/branches evolving under selection vs. neutrally in the context of previous literature’s estimates in the Discussion.

Regarding divergence in the context of other tissues/organs, we agree that this could be very interesting. However, because cross-species comparative RNA-seq analyses can be very sensitive to unevenness in the datasets generated and analyzed (e.g., depth, quality, replicates, the sexes

used, the species-specific annotations, and the reference genomes), we feel that these additional comparisons are best left for future analyses (after taking into consideration the above differences).

2. It would strengthen the paper to validate the results by comparing to previously published datasets. These studies are less comprehensive than the present study but would provide strong evidence for the quality of the present dataset.

We agree that the results depend on the quality of the underlying dataset and this is why we have invested extensively in the deep and uniform data generation across species and tissue, re-annotation, and the QC measures that we applied. As noted in the paper, we were very pleased with the average read depth and correlation between samples (“On average, we obtained 43 million mapped reads per sample with high correlations across triplicates (average Pearson correlation coefficient = 0.98”, pg. 3, and Methods). These values indicate deep and reproducible datasets across all samples. And though we referenced the repository that accompanies this project, we have now added emphasis that it contains QC files and plots on pg. 19 under RNA-seq read mapping:

“QC files and plots can be found within the project’s repository (<https://gitlab.com/EvoNeuro/sensory-rnaseq>), in particular see the `Full_Data_normPCA_trimmed.html` and `plot_mapping_stats_good_samples.html` files.”

We hope that this draws additional attention to the high quality of this dataset.

We note that we carried out extensive comparisons of detectable chemosensory receptor genes using all existing relevant datasets (summarized in Table S4). We found broad agreement between our data and published datasets, further indicating the high quality of the data.

3. One significant conclusion of the paper is that both cell type changes and expression within cell types contribute to gene expression evolution. However, the section where this is examined (p. 13 line 446) is referred to as preliminary. This conclusion should be strengthened if it is a major conclusion. One prediction is that if cell types are changing number, then most or all of the genes in the cell type would change expression in the same direction. If gene expression is changing within a cell type then the most coherent expression changes would likely be seen at the pathway level (examined in Fig. S3). Testing these predictions and differentiating between these modes of evolution would greatly strengthen the paper.

We agree that predicting the fraction of gene expression changes due to either cell composition changes or cell type-specific regulatory network modulations is a very exciting topic. However, we referred to this section as preliminary because of the challenge to directly address this question with the Fly Cell Atlas, which has only two replicates per sex for the tissues we examined. Single-cell RNA-seq generates highly variable cell proportions even between technical replicates and testing for cell composition differences would require more snRNA-seq replicates to statistically infer true cell proportion differences. Despite this, it made sense to us to analyze the data that exists. In addition, this section only focuses on the sex differences (not species differences) because the Fly Cell Atlas only contains data for *D. melanogaster*.

From a bulk RNA-seq perspective, we agree that the expansion of a cell type would result in higher gene expression for all genes in this cell type. However, the readout of a bulk RNA transcriptome is the average gene expression among all cell types composing the tissue and it would be very challenging to detect any cell type-specific pattern in the data. Cross-species single cell atlases are needed to address this question. Although highly relevant, we don’t think the data analyzed here could enable us to draw strong conclusions supporting one or the other hypothesis beyond the

preliminary sex differences that we report on. We have removed the cell composition claim from the abstract.

4. A downside of this study is the lack of connection between gene expression evolution in these chemosensory tissues and functional consequences. Figure 4 shows interesting shifts in chemosensory gene expression, but not any possible functions of these changes. This could be remedied by experiments to test the consequences or validation of previous studies with the present dataset. I.e. past studies that have found functional consequences of gene expression changes are captured by this dataset.

We agree that studying the functional evolution of these proteins would address many fascinating questions. We have used the results from this global study – along with the HCR experiments - to design additional experiments exactly along these lines. The generation of the transgenic reagents needed to do this work is however extensive and beyond the scope of the current manuscript.

Minor questions and comments:

1. The findings in Fig 1C and 1D for the ovipositor data seem to conflict. If *D. santomea* and *D. erecta* diverge at lower rates, why do they not cluster together more closely? Does this call into question the value of the analysis in 1C?

We note that for the ovipositor tree (Fig 1C) all bootstrap values strongly support the species clustering. And because the clustering analyses and relative rate tests are not measuring the exact same thing, some differences are expected. We agree that one expectation could be that the slowly evolving transcriptomes would have more similarities with each other than they do with transcriptomes that have diverged more quickly (both resembling more the ancestral state). However, the “speed” of divergence does not inform us about the directionality of gene expression changes. Two slowly evolving transcriptomes may still accumulate more differences with each other than with other species if they have been evolving in “opposite directions”.

2. Do the *de novo* gene annotations depend on genome quality?

Yes, *de novo* gene annotations do depend on genome assembly quality. For example, highly fragmented assemblies with many contigs and small scaffolds are likely to have genes missing (or fragmented) along with additional assembly errors. This can lead to uneven annotation and the possibility of propagating these biases in downstream analyses. Among our initial first steps when designing this experiment was to select species that had high-quality long-read assemblies to avoid such biases and to annotate all genomes the same way. Hence, this issue does not affect our specific study.

3. The paragraphs p. 5 line 196 and line 204 leave it unclear whether for these analyses divergence of a gene was considered in multiple or single species. This information should be specified in Figure 2 as well. It would be helpful to give some

The gene expression change analyses were performed across species. We have added “across species” to this section and modified the legend of Fig.2.

4. The analysis in Fig S5A seems to suggest that coincidental expression changes are inversely correlated with branch length. The authors should indicate whether they predict that these are resolved by later compensatory mutations to resolve pleiotropic mutations. This could be examined

by inferring coincidental changes in internal branches that were later resolved by compensatory mutations in a subset of later branches.

We performed a correlation between the number of shifts and species divergence time to verify that the number of shifts was not inflated on long branches (Fig. S6). We found no correlation, indicating that our shift estimates are not associated with branch length. Although we really like the idea of compensatory mutations accumulating over time to resolve former maladaptive expression changes, none of the correlations are significant and so we can't make any predictions on the topic.

5. P. 7 line 243 – “Our finding that most differentially expressed genes are broadly expressed” – it was not clear to me where this was already shown at this point in the paper. This statement seems contradicted by the statement in the next sentence that “genes that have changed in expression have similar modes of tissue breadth but tend to be more tissue-restricted than genes that have not changed.” More clarity would be helpful here.

The finding that differentially expressed genes are broadly expressed was shown in Fig 2B. In the section that the reviewer is asking about, we have raised the question whether differentially expressed genes are different in their breadth/specificity of expression compared to genes that were found to be unchanged in expression between species. When we made this comparison, there was not a difference in the mode of tissue breadth/specificity, but there is a difference in the right tail. This indicates that, while both classes of genes (differentially expressed and non-differentially expressed) are found to be expressed in most/many tissues, there is a tendency for differentially expressed genes to be limited to fewer tissues in comparison to genes that have not changed in expression. We have modified the several sentences of the section to add clarity.

6. The analysis in Figure 3 is a strength of the paper. In several places (e.g. p.7 line 246, 254) “breath” should be “breadth”. I also suggest switching tissue breadth to tissue specificity because the scale goes from 0 to 1 where 1 is the least breadth or greatest specificity. Alternatively, you could swap 0 and 1 to better reflect the measurement of breadth.

We recognize that this was unclear and also raised by reviewer 1. We have changed “breadth” to “specificity” throughout the manuscript.

7. The publication mentions that all of the genes chosen for functional validation confirmed the RNA-seq results except for Gr32a. How did Gr32a differ from the RNA-seq results?

As noted in our response to reviewer 2's question 8, we are uncertain why we were unable to detect *Gr32a* in *D. suzukii* antenna. Because we were successful in detecting *Gr32a* expression in the proboscis, we know that the probe works. One explanation might be that the tissues in the antenna of *D. suzukii* prevented sufficient penetration of the probes. We encountered this situation with several of *D. sechellia* tissues, which required specific optimization of our protocol (noted in the Methods section). While we unsuccessfully tried similar modifications to the protocol for *D. suzukii* antennae, perhaps there are successful parameter combinations that we have not tried.

The new Fig. S18 shows the results of the RNA-seq count data and the *Gr32a* HCR experiments in *D. suzukii*'s proboscis and antenna. And we have now added these details in the Results section (under *in situ* hybridization chain reaction experiments).

Reviewers' Comments:

Reviewer #1:

Remarks to the Author:

The authors have carefully addressed the concerns of all reviewers, further strengthening the quality of the manuscript. I believe this paper would be a valuable contribution to the field.

Reviewer #2:

Remarks to the Author:

Bontonou et al. have satisfactorily addressed the issues I raised in my previous review. I support the publication of the revised manuscript in Nature Communications.

Reviewer #3:

Remarks to the Author:

The authors have made extensive modifications to the manuscript including new analyses and clarifications. These changes address all of my concerns and suggestions. The authors should be commended on an excellent study.